# DiffAR: Denoising Diffusion Autoregressive Model for Raw Speech Waveform Generation

**Roi Benita[1] Michael Elad[1,2] Joseph Keshet[1]**
[1]Department of Electrical and Computer Engineering
[2]Department of Computer Science
Technion – Israel Institute of Technology, Haifa, Israel
`roibenita@campus.technion.ac.il`

## Abstract

Diffusion models have recently been shown to be relevant for high-quality speech generation. Most work has been focused on generating spectrograms, and as such, they further require a subsequent model to convert the spectrogram to a waveform (i.e., a vocoder). This work proposes a diffusion probabilistic end-to-end model for directly generating the raw speech waveform. The proposed model is autoregressive, generating overlapping frames sequentially, where each frame is conditioned on a portion of the previously generated one. Hence, our model can effectively synthesize an unlimited speech duration while preserving high-fidelity synthesis and temporal coherence. We implemented the proposed model for unconditional and conditional speech generation, where the latter can be driven by an input sequence of phonemes, amplitudes, and pitch values. Working directly on the waveform has some empirical advantages. Specifically, it allows the creation of local acoustic behaviors, like vocal fry, which makes the overall waveform sounds more natural. Furthermore, the proposed diffusion model is stochastic and not deterministic; therefore, each inference generates a slightly different waveform variation, enabling abundance of valid realizations. Experiments show that the proposed model generates speech with superior quality compared with other state-of-the-art neural speech generation systems. [1] [2]

## 1 Introduction

In the last two decades, impressive progress has been made in speech-based research and technologies. With these advancements, speech applications have become highly significant in communication and human-machine interactions. One aspect of this is generating high-quality, naturally-sounding synthetic speech, namely text-to-speech (TTS). In recent years, substantial research has been made to design a deep-learning-based generative audio model. Such an effective model can be used for speech generation, enhancement, denoising, and manipulation of audio signals.

Many neural-based generative models segment the synthesis process into two distinct components: a *decoder* and a *vocoder* (Zhang et al., 2023). The *decoder* takes a reference signal, like the intended text for synthetic production, and transforms it into acoustic features using intermediate representations, such as mel-spectrograms. The specific function of the decoder varies based on the application, which can be text-to-speech, image-to-speech, or speech-to-speech. The *vocoder*, on the other hand, receives these acoustic features and generates the associated waveform (Kong et al., 2020a). Although this two-step approach is widely adopted (Ren et al., 2020; Chen et al., 2020; Kim et al., 2020; Shen et al., 2018), one potential drawback is that focusing solely on the magnitude information (the spectrogram) might neglect certain natural and human perceptual qualities that can be derived from the phase (Oppenheim & Lim, 1981).

By contrast, *end-to-end* frameworks generate the waveform using a single model without explicitly producing acoustic features. The models *EATS* (Donahue et al., 2020), *Wave-Tacotron* (Weiss et al.,

---

[1]Code is available at `https://github.com/RBenita/DIFFAR`
[2]Audio samples are available at `https://rbenita.github.io/DIFFAR/`

2021), and *FastSpeech 2s* (Ren et al., 2020) pioneered efficient end-to-end training methods, but their synthesis quality lags behind two-stage systems. *VITS* (Kim et al., 2021) combines normalizing flow Rezende & Mohamed (2015) with VAE Kingma & Welling (2013) and adversarial training, achieving high-quality speech. *YourTTS* (Casanova et al., 2022) adopts an architecture similar to *VITS* and addresses zero-shot multi-speaker and multilingual tasks.

Recently, diffusion models have demonstrated impressive generative capabilities in synthesizing images, videos, and speech. In the context of speech synthesis, Numerous studies have suggested using diffusion models as decoders to generate the Mel-Spectrogram representation from a given text. *DiffTTS* (Jeong et al., 2021) leveraged the stochastic nature of diffusion models to capture a natural one-to-many relationship, allowing a given text input to be spoken in diverse ways. *Grad-TTS* (Popov et al., 2021), *ProDiff* (Huang et al., 2022b), *PriorGrad* (Lee et al., 2021), and *DiffGAN-TTS* (Liu et al., 2022) aimed to accelerate the synthesis process. However, it occasionally comes at the cost of audio quality, synthesis stochasticity, or model simplicity. *Guided-TTS* (Kim et al., 2022), functioning as a decoder, doesn't require a transcript of the target speaker, utilizing classifier guidance instead. In contrast, *DiffWave* (Kong et al., 2020b), *WaveGrad* (Chen et al., 2020) and *FastDiff* (Huang et al., 2022a) are vocoders that generate waveforms by conditioning the diffusion process on corresponding Mel-spectrograms. *DiffWave* (Kong et al., 2020b) can also learn the manifold of a limited, fixed-length vocabulary (the ten digits), producing consistent word-level pronunciations. It operates on the waveform directly but generates speech with a fixed duration of 1 second, meaning it cannot produce an entire sentence. Last, *WaveGrad 2* (Chen et al., 2021) is an end-to-end model consisting of (i) *Tacotron 2* (Elias et al., 2021) as an encoder for extracting an abstract hidden representation from a given phoneme sequence; and (ii) a decoder, which predicts the raw signal by refining the noisy waveform iteratively.

Large-scale generative models, such as *Voicebox* (Le et al., 2023) and *VALL-E* (Wang et al., 2023), achieve state-of-the-art results by leveraging extensive audio datasets. *Voicebox*, functioning as an acoustic infilling model, and *VALL-E* as an end-to-end model which utilizing *Codec* representation (Défossez et al., 2022), excel in simulating natural phenomena and enabling in-context learning. These models demonstrate robustness to noise and the ability to generalize when trained on thousands hours of speech. However, replicating this success on smaller datasets presents a significant challenge.

Parallel to that, the generation of long waveforms can be effectively achieved using an autoregressive (AR) approach. This involves the sequential generation of waveform samples during the inference phase (e.g., Oord et al., 2016; Wang et al., 2023). While autoregressive models work well for TTS, their inference is slow due to their sequential nature. On the other hand, non-autoregressive models such as (Ren et al., 2020; Chen et al., 2021) struggle to generate extremely long audio clips that correspond to a long text sequence due to the limited GPU memory.

This work proposes a novel autoregressive diffusion model for generating raw audio waveforms by sequentially producing short overlapping frames. Our model is called *DiffAR* – Denoising Diffusion Autoregressive Model. It can operate in an unconditional mode, where no text is provided, or in a conditional mode, where text and other linguistic parameters are used as input. Because our model is autoregressive, it can generate signals of an arbitrary duration, unlike *DiffWave*. This allows the model to preserve coherent temporal dependencies and maintain critical characteristics. *DiffAR* is an end-to-end model that directly works on the raw waveform *without* any intermediate representation such as the Mel-spectrogram. By considering both the amplitude and phase components, it can generate a reliable and human-like voice that contains everyday speech phenomena including *vocal fry*, which refers to a voice quality characterized by irregular glottal opening and low pitch, and often used in American English to mark phrase finality, sociolinguistic factors and affect.

We are not the first to introduce autoregressive diffusion models. Ho et al. (2022) proposed a method for video synthesis, and Hoogeboom et al. (2021) extended diffusion models to handle ordered structures while aiming to enhance efficiency in the process. Our model focuses on one-dimensional time-sequential data, particularly unlimited-duration high-quality speech generation.

The contributions of the paper are as follows: (i) An autoregressive denoising diffusion model for high-quality speech synthesis; (ii) This model can generate unlimited waveform durations while preserving the computational resources; and (iii) This model generates human-like voice, including vocal fry, with a high speech quality and variability compared to other state-of-the-art models.

This paper is organized as follows. In Section 2, we formulate the problem and present our autoregressive approach to the diffusion process for speech synthesis. Our model, *DiffAR*, can be conditioned on input text, described in Section 3. In Section 4 we detail *DiffAR*'s architecture. Next, in Section 5, we present the empirical results, including a comparison to other methods and an ablation study. We conclude the paper in Section 6.

## 2 PROPOSED MODEL

Our goal is to generate a speech waveform that mimics the human voice and sounds natural. We denote the waveform by $\mathbf{x} = (x_1, \ldots, x_T)$ where each sample $x_t \in [-1, 1]$. The number of samples, $T$, is not fixed and varies between waveforms. To do so, we consider the joint probability distribution of the speech $p(\mathbf{x})$ from a training set of speech examples, $\{\mathbf{x}_i\}_{i=1}^{N}$. Each sample from this distribution would generate a new valid waveform. This is the *unconditional* case.

Our ultimate objective is to generate the speech from a specified text. To convert text into speech, we specify the text using its linguistic and phonetic representation $\mathbf{y} = (y_1, \ldots, y_T)$, where we can consider $y_t$ to be the phoneme at time $t$, and may also include the energy, pitch or other temporal linguistic data. In the *conditional* case we estimate the conditional distribution $p(\mathbf{x}|\mathbf{y})$ from the transcribed training set $\{\mathbf{x}_i, \mathbf{y}_i\}_{i=1}^{N}$. Sampling from this distribution generates a speech of the input text given by $\mathbf{y}$.

To generate a waveform of an arbitrary length $T$, our model operates in frames, each containing a fixed $L$ samples, where $L \ll T$. Let $\mathbf{x}^l$ denote a vector of samples representing the $l$-th frame. To ensure a seamless transition between consecutive frames, we *overlap* them by shifting the starting position by $L_o$ samples. We propose an autoregressive model wherein the generation of the current frame $l$ is conditioned on the last $L_o$ samples of the previous frame $l-1$. See Figure 1.

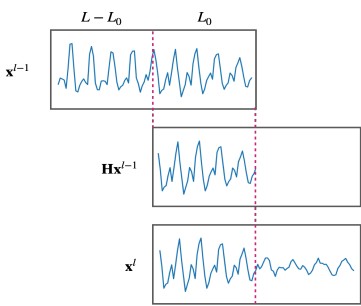

Following these definitions, we assume Markovian conditional independence and denote the probability distribution of the $l$-th speech frame by $p(\mathbf{x}^l|\mathbf{x}^{l-1})$, indicating that it is dependent on the preceding frame, $l-1$, but not conditioned on the frames before that or on any input text (unconditional case). Similarly, let $p(\mathbf{x}^l|\mathbf{x}^{l-1}, \mathbf{y}^l)$ be the probability distribution conditioned also on a specified input text. The sequence $\mathbf{y}^l$ stands for the linguistic-phonetic representation of the $l$-th frame, which will be discussed in the following section.

Figure 1: The autoregressive model uses part of the previous frame to generate the current frame.

Our approach is based on denoising diffusion probabilistic models (DDPM; Ho et al., 2020). A diffusion model is a generative procedure involving latent variables constructed from two stochastic processes: the *forward* and the *reverse* processes. Each process is defined as a fixed *Markovian* chain composed of $S$ latent instances of the $l$-th speech frame $\mathbf{x}_1^l, \ldots, \mathbf{x}_S^l$. We denote the source speech frame as the 0-th process element, $\mathbf{x}_0^l = \mathbf{x}^l$.

During the *forward process*, a small Gaussian noise is gradually mixed with the original speech frame $\mathbf{x}_0^l$ through $S$ diffusion steps. The step sizes are predefined by a variance schedule $\{\beta_t \in (0, 1)\}_{s=1}^{S}$, which gradually transforms the original frame $\mathbf{x}_0^l$ into the last latent variable $\mathbf{x}_S^l$ that follows an isotropic Gaussian distribution $\mathbf{x}_S^l \sim \mathcal{N}(0, \mathbf{I}_L)$. Denote by $q$ the distribution of the forward process, and taking into account its Markovian nature, we have:

$$q\left(\mathbf{x}_{1:S}^l|\mathbf{x}_0^l\right) = \prod_{s=1}^{S} q(\mathbf{x}_s^l|\mathbf{x}_{s-1}^l) . \tag{1}$$

Following Ho et al. (2020), the conditional process distribution $q$ is parameterized as Gaussian distribution as follows:

$$q(\mathbf{x}_s^l|\mathbf{x}_{s-1}^l) = \mathcal{N}\left(\mathbf{x}_s^l; \sqrt{1-\beta_s}\mathbf{x}_{s-1}^l, \beta_s\mathbf{I}_L\right) . \tag{2}$$

Note that the distribution $q$ is not directly influenced by the previous frame $\mathbf{x}^{l-1}$, nor by the input text $\mathbf{y}^l$ in the conditional case.

The *reverse process* aims to recover the original speech frame $\mathbf{x}_0^l$ from the corrupted frame $\mathbf{x}_S^l$ by progressively denoises it. The probability distribution of the reverse process takes into account the autoregressive property of our overall model, conditioned on the previous frame $\mathbf{x}^{l-1}$ and the input text if given. The reverse process, also under the Markovian assumption, is defined as the conditional distribution:

$$p_\theta\left(\mathbf{x}_{0:S}^l \mid \mathbf{x}^{l-1}, \mathbf{y}^l\right) = p(\mathbf{x}_S^l) \prod_{s=1}^S p_\theta(\mathbf{x}_{s-1}^l \mid \mathbf{x}_s^l, \mathbf{x}^{l-1}, \mathbf{y}^l), \tag{3}$$

where $p_\theta$ is a learned model with parameters $\theta$, and $\mathbf{y}^l$ is either given in the conditional case or omitted in the unconditional case. To be precise, the learned model uses the overlap portion of the previous frame, namely $L_o$ samples. We use the notation $\mathbf{H}\mathbf{x}^{l-1}$ to specify the overlapped segment of the previous frame (Figure 1), where $\mathbf{H} \in \mathbb{R}^{L \times L}$ is an *inpainting* and *reordering* matrix, which is defined as follows:

$$\mathbf{H} = \left[ \begin{array}{cc} \mathbf{0} & \mathbf{I}_{L_o} \\ \mathbf{0} & \mathbf{0} \end{array} \right]. \tag{4}$$

Beyond the Markovain factorization, as shown above, we further assume that each transition for a time step $s$ is represented as drawn from a Gaussian distribution:

$$p_\theta(\mathbf{x}_{s-1}^l \mid \mathbf{x}_s^l, \mathbf{x}^{l-1}, \mathbf{y}^l) = \mathcal{N}\left(\mathbf{x}_{s-1}^l; \ \mu_\theta\left(\mathbf{x}_s^l, \mathbf{H}\mathbf{x}^{l-1}, \mathbf{y}^l, s\right), \Sigma_\theta\left(\mathbf{x}_s^l, \mathbf{H}\mathbf{x}^{l-1}, \mathbf{y}^l, s\right)\right). \tag{5}$$

Training is performed by minimizing the variational bound on the negative log-likelihood while using the property that relates $\mathbf{x}_s^l$ directly with $\mathbf{x}_0^l$ (Ho et al., 2020):

$$\mathbf{x}_s^l = \sqrt{\bar{\alpha}_s}\mathbf{x}_0^l + \sqrt{1 - \bar{\alpha}_s}\boldsymbol{\epsilon}_s \quad \boldsymbol{\epsilon}_s \sim \mathcal{N}(\mathbf{0}, \mathbf{I}), \tag{6}$$

where $\alpha_s = 1 - \beta_s$, $\bar{\alpha}_s = \prod_{i=1}^s \alpha_i$. The loss is reduced as follows:

$$\mathcal{L}_s = \mathbb{E}_{\mathbf{x}_0^l, \boldsymbol{\epsilon}_s}\left[\left\|\boldsymbol{\epsilon}_\theta\left(\sqrt{\bar{\alpha}_s}\mathbf{x}_0^l + \sqrt{1 - \bar{\alpha}_s}\boldsymbol{\epsilon}_s, \mathbf{H}\mathbf{x}^{l-1}, \mathbf{y}^l, s\right) - \boldsymbol{\epsilon}_s\right\|^2\right], \tag{7}$$

where $\boldsymbol{\epsilon}_\theta$ is an approximation of $\boldsymbol{\epsilon}_s$ from $\mathbf{x}_s$ with parameters $\theta$ and $s$ is uniformly taken from the entire set of diffusion time-steps. In summary, our model aims to learn the function $\boldsymbol{\epsilon}_\theta$, which acts as a conditional *denoiser*. This function can be used along with a noisy speech frame to estimate a clean version of it.

The *inference* procedure is sequential and carried out autoregressively for each frame. Assume we would like to generate the $l$-th frame, given the already generated previous frame $\hat{\mathbf{x}}^{l-1}$. For the new frame generation we apply the following equation iteratively from $s = S$:

$$\mathbf{x}_{s-1}^l = \frac{1}{\sqrt{\alpha_s}}\left(\mathbf{x}_s^l - \frac{1 - \alpha_s}{\sqrt{1 - \bar{\alpha}_s}}\boldsymbol{\epsilon}_\theta\left(\mathbf{x}_s^l, \mathbf{H}\hat{\mathbf{x}}^{l-1}, \mathbf{y}^l, s\right)\right) + \sigma_s\mathbf{z}_s, \tag{8}$$

where $\boldsymbol{\epsilon}_\theta$ is the learned model, $\mathbf{z}_s \sim \mathcal{N}(\mathbf{0}, \mathbf{I})$ and $\sigma_s = \sqrt{\frac{1 - \bar{\alpha}_{s-1}}{1 - \bar{\alpha}_s}\beta_s}$. To initiate the generation, we designate the initial frame ($l = 0$) as a silence one. In the last iteration, when $s = 1$, we use $\mathbf{z}_1 = \mathbf{0}$.

## 3 TEXT REPRESENTATION AS LINGUISTIC AND PHONOLOGICAL UNITS

Recall that our ultimate goal is to synthesize speech given an input text. Following Ren et al. (2020); Kim et al. (2020); Chen et al. (2021), we use the phonetic representation of the desirable text as a conditioned embedding, as it accurately describes how the speech should be produced. Let $\mathcal{Y}$ represent the set of phonemes, $|\mathcal{Y}| = 72$. Recall that in our setup, we are required to supply the phonetic content for each frame, denoted as $\mathbf{y}^l$. This entails a vector comprising $L$ values, where each value represents a phoneme from the set $\mathcal{Y}$ for each respective sample. Note that while the phoneme change rate is much slower than the sampling frequency, we found this notation clearer for our discussion.

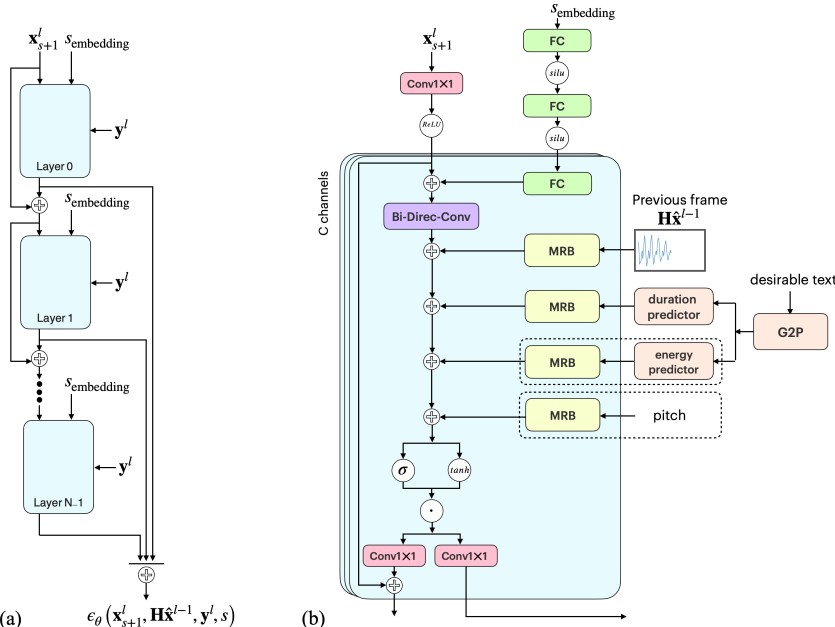

Figure 2: (a) A general overview of the structure of the residual layers and their interconnections. (b) A detailed overview of a single residual layer.

Since the actual text is given as a sequence of words, during training, we transform the text sequence into phonemes and their corresponding timed phoneme sequence using a phoneme alignment procedure. This process identifies the time span of each phoneme within the waveform (McAuliffe et al., 2017). During inference, we do not have a waveform as our goal is to generate one, and we use a *grapheme-to-phoneme* (G2P) component to convert the words into phonemes (Park & Kim, 2019) and a *duration predictor* to estimate the time of each phoneme.

**Duration Predictor.** The duration prediction is a small neural network that gets as input a phoneme and outputs its typical duration. The implementation details are given in Appendix (C.1). During inference, the generated frame duration is allowed to deviate from the exact value $L$, since we restrict the vector $\mathbf{y}^l$ to encompass entire phoneme time-spans, which is easier to manage.

The speech corresponding to each text can be expressed in various ways, particularly when the transition is executed directly on the waveform. Utilizing a diffusion process to implement the model further amplifies this variability, owing to the stochastic nature of the process. On the one hand, we aim to retain the diversity generated by the model to facilitate more reliable and nuanced speech. On the other hand, we aspire to steer and regulate the process to achieve natural-sounding speech. Consequently, following the approach outlined in Ren et al. (2020), we allow the incorporation of elements such as energy and pitch predictors into the process. Namely we enhance the vector $\mathbf{y}^l$ to include other linguistic information rather than just phonemes.

**Energy Predictor.** We gain significant flexibility and control over the resulting waveform by directly conditioning our model on the energy signal. Instead of relying on the estimated output of the energy predictor, we have the autonomy to determine the perceived loudness of each phoneme. Our approach offers guidance to the synthesis process while still governed by the inherent stochasticity of the diffusion model. Much like the duration predictor, our energy predictor was trained to predict the relative energy associated with each phoneme. Detailed information about the implementation of the energy predictor can be found in Appendix C.2.

**Pitch.** Pitch, or fundamental frequency, is another critical element in the structure of the waveform. To assess its impact on the synthesis process while conditioned on a given pitch contour, we also decided to incorporate it into our model. In this case, we did not build a pitch predictor and used a given sequence of pitch values, estimated using state-of-the-art method (Segal et al., 2021).

## 4 MODEL ARCHITECTURE

The architecture of *DiffAR* is shown in the Figure 2. The model's backbone is based on the *Dif-fWave* architecture (Kong et al., 2020b). Figure 2(a) illustrates the general structure of the network. The network consists of $N = 36$ residual layers, each containing $C = 256$ residual channels. The output from each layer is integrated with the accumulated outputs from previous ones. These combined outputs are fed into a network of two fully connected layers, which leverage ReLU activation functions to generate the final output. The layer dimensions are described in the Appendix C.

Figure 2(b) schematically depicts a single channel of the residual layer. This layer employs the bidirectional dilated convolution architecture (Oord et al., 2016), which facilitates parallel inference for each frame through a dilation cycle of $[1, 2, \ldots, 2048]$. To foster an autoregressive progression, the layer is conditioned on $\mathbf{H}\mathbf{x}^{l-1}$, incorporating essential information from the previous frame. The indication of the diffusion step $s$ is accomplished by employing a 128-dimensional encoding vector for each $s$ (Vaswani et al., 2017) as input to the model, similar to the approach used in (Kong et al., 2020b). Additionally, *DiffAR* can be conditioned on optional data, including the targeted phonemes, the desired energy, and the desired pitch.

Each conditioned signal passes through a Multi-scaled Residual Block (MRB) and is then summed to the output of the bidirectional convolutional component. The MRBs comprise three convolutional layers with kernels $[3, 5, 7]$ and use the identical dilation pattern as the residual layer. These MRBs are trained concurrently with the model.

## 5 EXPERIMENTS

In this section, we comprehensively evaluate our model through empirical analysis. Initially, we explore unconditional speech generation, wherein a specific text does not constrain the synthesis. Subsequently, we discuss our conditional model, employed when there is a designated text to synthesize. We compare our model with two TTS models: *WaveGrad 2* (Chen et al., 2021) and *Fast-Speech 2* (Ren et al., 2020). We then turn to a short ablation study, comparing different parts of our model. Furthermore, in Appendix A we present our model stochasticity and controllability, while in Appendix B we present the synthesis of vocal fry, which is unique to our model. Lastly, a comprehensive comparison with various acoustic and end-to-end models is provided in Appendix D

All models were trained and evaluated on the LJ-Speech (Ito & Johnson, 2017) dataset, which consists of 13,100 short audio clips (about 24 hours) of a female speaker. The dataset was divided into three subsets: 12,838 samples for the training set, 131 samples for the test set, and an additional 131 samples for the validation set. Throughout the experiments, we maintained the original LJ-Speech data partitioning.

In all the experiments, we used relatively long frame durations (e.g., $L = 500$ and $L_o = 250$ milliseconds). We would like to point out that a conventional frame length of 20-30 milliseconds and a shift of 10 milliseconds, often used in speech processing, are established based on the stationary properties of speech. However, this is not a concern in diffusion models, thereby permitting us to employ substantially larger frame sizes. This aids the diffusion process in seamlessly modeling the waveform encompassing three-four consecutive phonemes in the newly generated segment.

### 5.1 UNCONDITIONAL SPEECH GENERATION

First, we created a model entirely unconditioned by external factors, relying solely on information from the previous frame. The main goal is to assess whether generating a sequence of frames, as outlined in the autoregressive approach in Section 2, results in a continuous signal with seamless transitions.

During the training phase, we fixed the frame length settings to $(L, L_o) = (1000, 500)$, utilizing $S = 200$ diffusion steps. We utilize a noise schedule parameterized by $\beta_t \in [1 \times 10^{-4}, 0.02]$ to control the diffusion process. However, in the synthesis phase, we assessed the model's ability to generalize across different frame lengths, specifically considering $(L, L_o) = \{(1000, 500), (500, 250), (400, 200)\}$. Examples can be found in our model's GitHub repository.

Table 1: Comparison to WaveGrad 2 (Chen et al., 2021)

| Method | ↑MOS | ↑MOS scaled | ↑MUSHRA | ↓CER(%) | ↓WER(%) |
|---|---|---|---|---|---|
| Ground truth | $3.98 \pm 0.08$ | $4.70 \pm 0.09$ | $71.2 \pm 2.0$ | 0.89 | 2.13 |
| WaveGrad 2 | $3.61 \pm 0.09$ | $4.26 \pm 0.10$ | $63.8 \pm 2.3$ | 3.47 | 5.75 |
| DiffAR (200 steps) | $3.75 \pm 0.08$ | $4.43 \pm 0.10$ | $65.7 \pm 2.2$ | 2.67 | 6.16 |
| DiffAR (1000 steps) | $3.77 \pm 0.08$ | $\mathbf{4.45 \pm 0.09}$ | $\mathbf{66.7 \pm 2.2}$ | **1.95** | **4.65** |

The generated signals exhibit smooth transitions and connectivity, indicating that the *DiffAR* architecture has effectively learned local dependencies. However, the model generated non-existent but human language-like words (similar to Oord et al., 2016; Weiss et al., 2021). Additionally, we observed that global dependencies are improved as the frame length increases, utilizing the entire learned receptive field. This result is not unexpected, considering the model does not condition on textual information. Modeling a manifold that generates a large vocabulary and meaningful words without textual guidance is still challenging. On the other hand, a simple manifold for only ten digits can be successfully generated (Kong et al., 2020b).

## 5.2 CONDITIONAL SPEECH GENERATION

We conducted a comparative study of our conditional model against other TTS models. Although there is a plethora of TTS systems available, our objective was to benchmark against high-performing and relevant models *WaveGrad 2* and *FastSpeech 2*.[3]

We evaluated the synthesized speech using two subjective and two objective metrics. For subjective measurement, we used the mean opinion scores (MOS), where 45 samples from the test set are evaluated for each system, and 10 ratings are collected for each sample. Raters were recruited using the Amazon Mechanical Turk platform, and they were asked to evaluate the quality of the speech on a scale of 1 to 5. Despite their advantages, MOS tests can be challenging to compare between different papers (Kirkland et al., 2023), and they may even exhibit bias within the same study, due to the influence of samples from other systems in the same trial.

To mitigate these challenges and provide a more robust evaluation framework, we used another subjective evaluation – the Multiple Stimuli with Hidden Reference and Anchor (MUSHRA) test. We followed the MUSHRA protocol (Series, 2014), using both a hidden reference and a low anchor. For the overall quality test, raters were asked to rate the perceptual quality of the provided samples from 1 to 100. We report average ratings along with a $95\%$ confidence interval for both metrics.

We randomly selected 60 recordings from the test set for an objective assessment and used their text to re-synthesize waveforms. We evaluated the generated waveforms using state-of-the-art automatic speech recognition (Whisper medium model; Radford et al., 2023) and reported the character error rate (CER) and the word error rate (WER) relative to the original text.

During the training phase, we fixed the frame length settings to $(L, L_o) = (500, 250)$. We utilize a noise schedule $\beta_t \in \left[1 \times 10^{-4}, 0.02\right]$. We trained two models – one with $S = 200$ steps and one with 1000 steps. During inference, the models were conditioned on phonemes (obtained from G2P unit (Park & Kim, 2019)), the predicted durations, and the predicted energy.

**WaveGrad 2.** We start by describing a comparison of our model to *WaveGrad 2* (Chen et al., 2021), which is an encoder-decoder end-to-end waveform generation system that is based on diffusion models. We used an unofficial implementation[4] of it as the original one is unavailable. Results for *WaveGrad 2* are presented in Table 1. Each row represents a different model, where the first row, denoted *Ground truth*, represents the performance with the original waveforms from the database, and it is given as a reference. For each model we show the results of MOS, MUSHRA, CER and WER. The column labeled **MOS scaled** indicates the adjusted MOS results, which have been scaled proportionately to align with the MOS values of ground truth and *WaveGrad 2* (Chen et al., 2021).

---

[3]A comprehensive comparison with various publicly available acoustic and end-to-end models, is provided in Appendix D

[4]https://github.com/maum-ai/wavegrad2

Table 2: Comparison to FastSpeech 2 (Ren et al., 2020)

| Method | ↑MOS | ↑MOS scaled | ↑MUSHRA | ↓CER(%) | ↓WER(%) |
|---|---|---|---|---|---|
| Ground truth | $3.98 \pm 0.05$ | $4.22 \pm 0.05$ | $68.9 \pm 1.4$ | 0.89 | 2.13 |
| FastSpeech 2 | $3.54 \pm 0.09$ | $3.75 \pm 0.09$ | $63.4 \pm 2.2$ | 2.15 | 4.82 |
| FastSpeech 2 improved | $3.75 \pm 0.08$ | $3.98 \pm 0.09$ | $64.2 \pm 2.5$ | **1.73** | **4.31** |
| DiffAR (200 steps) | $3.76 \pm 0.05$ | $3.99 \pm 0.06$ | $66.0 \pm 1.5$ | 2.67 | 6.16 |
| DiffAR (1000 steps) | $3.82 \pm 0.05$ | $\mathbf{4.05 \pm 0.06}$ | $\mathbf{66.2 \pm 1.5}$ | 1.95 | 4.65 |

The table illustrates that our model surpasses *WaveGrad 2* across all evaluated metrics. This can be attributable to the fact that *WaveGrad 2* uses an architecture that generates the entire utterance in a single instance instead of operating in an autoregressive manner like *DiffAR*.

**FastSpeech 2.** We turn now to compare *DiffAR* with *FastSpeech 2* (Ren et al., 2020), as non-diffusion acoustic model, implementing the two-state decoder-vocoder approach. We used an unofficial implementation[5] as the original one associated with the paper was not made available. We evaluated two versions of this model: the original *FastSpeech 2*, as described in Ren et al. (2020), and an improved version, which uses additional *Tacotron-2* Shen et al. (2018) style post-net after the decoder, gradient clipping during the training, phoneme-level pitch and energy prediction instead of frame-level prediction, and normalizing the pitch and energy features.[6] Both versions were trained on the LJ-speech dataset, with a pre-trained HiFi-GAN (Chu et al., 2017) as a vocoder. The results are given in Table 2. Like the previous table, the rows represent different models, and the columns are the evaluation metrics. It is important to note that the subjective evaluation (MOS and MUSHRA tests) were carried out independently for *WaveGrad 2* and *FastSpeech 2* to ensure the results were not influenced by each other. Also note that the column **MOS scaled** in this table was scaled proportionally to the ground-truth and *FastSpeech 2* MOS values, as reported in Ren et al. (2020).

Based on the MOS and MUSHRA values, it is evident that our model generates speech characterized by higher quality and a more natural sound, compared to *FastSpeech 2*, and in the same ballpark compared to *FastSpeech 2-improved*. By analyzing the CER and WER values, it is evident that our model achieves slightly greater intelligibility than *FastSpeech 2*, yet still falls short of the performance of *FastSpeech 2-improved*.

A comprehensive comparison with various acoustic and end-to-end models, including both diffusion-based and non-diffusion-based approaches, is provided in Appendix D. We provide a comparison of *DiffAR* to these models in terms of audio quality, the *one-to-many* diverse speech realizations for a given text, and the simplicity of its architecture. Additionally, the synthesis time factor is addressed in Appendix E.

### 5.3 ABLATION STUDY

In this section, we introduce an ablation study designed to evaluate the impact of integrating additional components into the model and assess these components' contribution to the observed error rates. We carried out evaluations based on CER and WER metrics to accomplish this.

The results are presented in Table 3. The table is structured to present the ablation results initially when the conditioning is based on the ground truth values for linguistic and phonetic content. Subsequently, we showcase the ablation results obtained with predicted values. The *DiffAR-E* model denotes a variant conditioned on phonemes and their respective durations but not on the energy. In contrast, the *DiffAR* model is conditioned on phonemes, their durations, and their energy levels. Lastly, the *DiffAR+P* model represents a version that additionally incorporates pitch conditioning. The number in parentheses indicates the number of diffusion steps each model was trained and

---

[5] https://github.com/ming024/FastSpeech2.

[6] Ideally, we should have also compared our model to *FastSpeech 2s* (Ren et al., 2020), which an is end-to-end text-to-waveform system, and to *Wave-Tacotron* (Weiss et al., 2021), but no implementations have been found for these models.

Table 3: Intelligibility of different configurations of *DiffAR*, where different phonetic and linguistic values are either true or predicted.

| Method | Phonemes | Durations | Energy | Pitch | ↓CER(%) | ↓WER(%) |
|---|---|---|---|---|---|---|
| Ground truth | – | – | – | – | 0.89 | 2.13 |
| DiffAR-E (200) | true | true | – | – | 2.90 | 5.98 |
| DiffAR (200) | true | true | true | – | 1.18 | 3.96 |
| DiffAR (1000) | true | true | true | – | 1.70 | 4.25 |
| DiffAR+P (200) | true | true | true | true | 1.12 | 3.47 |
| DiffAR-E (200) | true | pred | – | – | 2.68 | 6.09 |
| DiffAR-E (200) | pred | pred | – | – | 3.35 | 7.41 |
| DiffAR (200) | true | pred | pred | – | 1.05 | 3.09 |
| DiffAR (1000) | true | pred | pred | – | 2.03 | 4.34 |
| DiffAR (200) | pred | pred | pred | – | 2.67 | 6.16 |
| DiffAR (1000) | pred | pred | pred | – | 1.95 | 4.65 |

tested. The first set of columns indicates whether the model was conditioned on true or predicted values. The final two columns provide the CER and WER values.

It can be seen from the results that as we incorporate more supplementary information into the process, the quality of the results improves. In addition, as the task approaches a more realistic scenario, where the only source of original information is the text itself, we observe an increase in values, and inaccuracies appear to be linked to the prediction components. Nevertheless, it is noteworthy that by increasing the number of diffusion steps in the process, the model seems capable of autonomously learning crucial relationships, resulting in lower error values in the realistic scenario compared to a shorter process. Another notable finding is that when we have access to the original energy and pitch information, we achieve results that closely approximate ground truth. This outcome is expected, as this information plays a significant role in modeling the characteristics of a natural waveform signal.

Another noteworthy aspect highlighted in these results is the balance between the inherent stochasticity of the diffusion process and the degree of controllability achieved through conditioning the model with supplementary information. A more detailed demonstration is provided in Appendix A.

## 6 CONCLUSION

In this work, we proposed *DiffAR*, an end-to-end denoising diffusion autoregressive model designed to address audio synthesis tasks, specifically focusing on TTS applications. Our model incorporates a carefully selected set of characteristics, each contributing significantly to its overall performance. The diffusive process enriches synthesis quality and introduces stochasticity, while the autoregressive nature enables the handling of temporal signals without temporal constraints and facilitates effective integration with the diffusive process. Synthesizing the waveform directly, without using any intermediate representations enhanced the variability and simplified the training procedure. By estimating both the phase and amplitude, *DiffAR* enables the modeling of phenomena such as vocal fry phonation, resulting in more natural-sounding signals. Furthermore, The architecture of *DiffAR* offers simplicity and versatility, providing explicit control over the output signal. These characteristics are interconnected, and their synergy contributes to the model's ability to outperform leading models in terms of both intelligibility and audio quality.

Like other autoregressive models, *DiffAR* model faces the challenge of long synthesis times. Future work can focus on reducing synthesis time by using fewer diffusion steps (Song et al., 2020) or by exploring methods to expedite the process (Hoogeboom et al., 2021). Another avenue for improvement is conditioning the model with elements like speaker identity and emotions and incorporating classifier-free guidance (Ho & Salimans, 2021) to handle such various conditions effectively.[7] Lastly, ablation studies suggest that enhancing the force aligner, grapheme-to-phoneme, and prediction components could significantly improve the results.

---

[7]A more detailed discussion is provided in Appendix F.

## 7 REPRODUCIBILITY

To ensure the work is as reproducible as possible and comparable with future models, we have provided comprehensive descriptions of both the training and the sampling procedures. The main ideas of the method are presented in section 2. The model architecture is provided in Section 4 and is also presented in a more detailed format in Appendix C. In addition, our complete code for training and inference, along with hyperparameter settings to run experiments, can be found under the project's GitHub repository `https://github.com/RBenita/DIFFAR`. Audio samples are available at `https://rbenita.github.io/DIFFAR/`.

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

APPENDIX

## A  STOCHASTICITY AND CONTROLLABILITY THROUGH THE GENERATIVE PROCESS

The inherent stochasticity within the diffusion process, particularly when it models the raw waveform itself, enables creative synthesis with a substantial degree of freedom in terms of energy, pitch, and timing. This variability is a crucial element within synthesis models as it contributes uniqueness and distinctiveness to the generated signal. One valuable application of such a model is its potential utility for augmenting speech signals.

Conditioning the synthesis process with supplementary information, such as pitch, energy, or phonemes, enables extensive control over the generated output. Using that information steers the synthesis procedure towards more precise regions within the manifold. This, in turn, leads to the generation of signals that exhibit a higher degree of desired and shared characteristics.

One notable advantage of the *DiffAR* model is its capability to effectively balance the trade-off between stochasticity and controllability within the synthesis process. On one hand, it operates as an end-to-end model based on diffusion principles, amplifying the process's inherent variability. On the other hand, it offers a versatile architecture that enables explicit conditioning of desirable information. By doing that, it provides the model with more context and specific guidance.

Figure 3 illustrates the trade-off between controllability and stochasticity, as demonstrated in five different models. For each model, we compared the energy and pitch among five syntheses of the same text. Figure 3a illustrates the outcomes of *DiffAR-E*. As this model is exclusively conditioned on the phonemes and their durations, its signals demonstrate significant pitch and energy variation. Figure 3b illustrates the conditioning of the process on a desired energy level. When utilizing the *DiffAR (200)* model, the generated signals tend to have energy values that are quite similar, showing slight variation around the desired values (indicated by the red line). However, there is still a significant variability in the pitch values across the generated signals. Figure 3c illustrates five signals generated by the *DiffAR+P* model, which also conditions the synthesis process on the desired pitch values. In this case, this conditioning significantly diminishes the variability among the generated signals. The pitch and energy values of all signals become remarkably similar and closely resemble the values of the conditioned inputs. It is important to note that due to the use of diffusion and the high variability in the raw waveform itself, the variation still exists, and the audio clips are not completely identical.

In contrast to our model, which provides extensive control over the signal properties and variability, Figure 3d illustrates that *FastSpeech 2* lacks any variability. When given a specific text input, all corresponding syntheses exhibit uniform pitch and energy characteristics, resulting in identical signals. On the other hand, the *WaveGrad 2* model does introduce variability, as depicted in Figure 3e, and this variability can be adjusted by reducing the number of diffusion steps. However, it's worth noting that both *WaveGrad 2* and *FastSpeech 2* lack the capability to explicitly manipulate the signal towards predefined energy or pitch target values.

*Vocal fry*, also known as *creaky-voice*, is a vocal phenomenon characterized by a low and scratchy sound that occupies the vocal range below modal voice. Recently, vocal fry has gained popularity in various areas, including the United States, and is observed in both women and men. This type of production can signal the end of an utterance but even as a sociolinguistic marker for distinguishing a speech group from another within the same language.

The LJ-Speech dataset contains numerous segments featuring vocal fry. These portions are typically distinguished by their low and irregular fundamental frequency ($F0$), reduced energy, and damped pulses (Keating et al., 2015). An example can be found in Figure 4a.

## B  VOCAL FRY

Modeling vocal fry behavior in synthesis applications presents a non-trivial challenge that researchers have previously attempted to address (Narendra & Rao, 2017). The complexity arises because a significant portion of the relevant information is embedded in the phase component of the

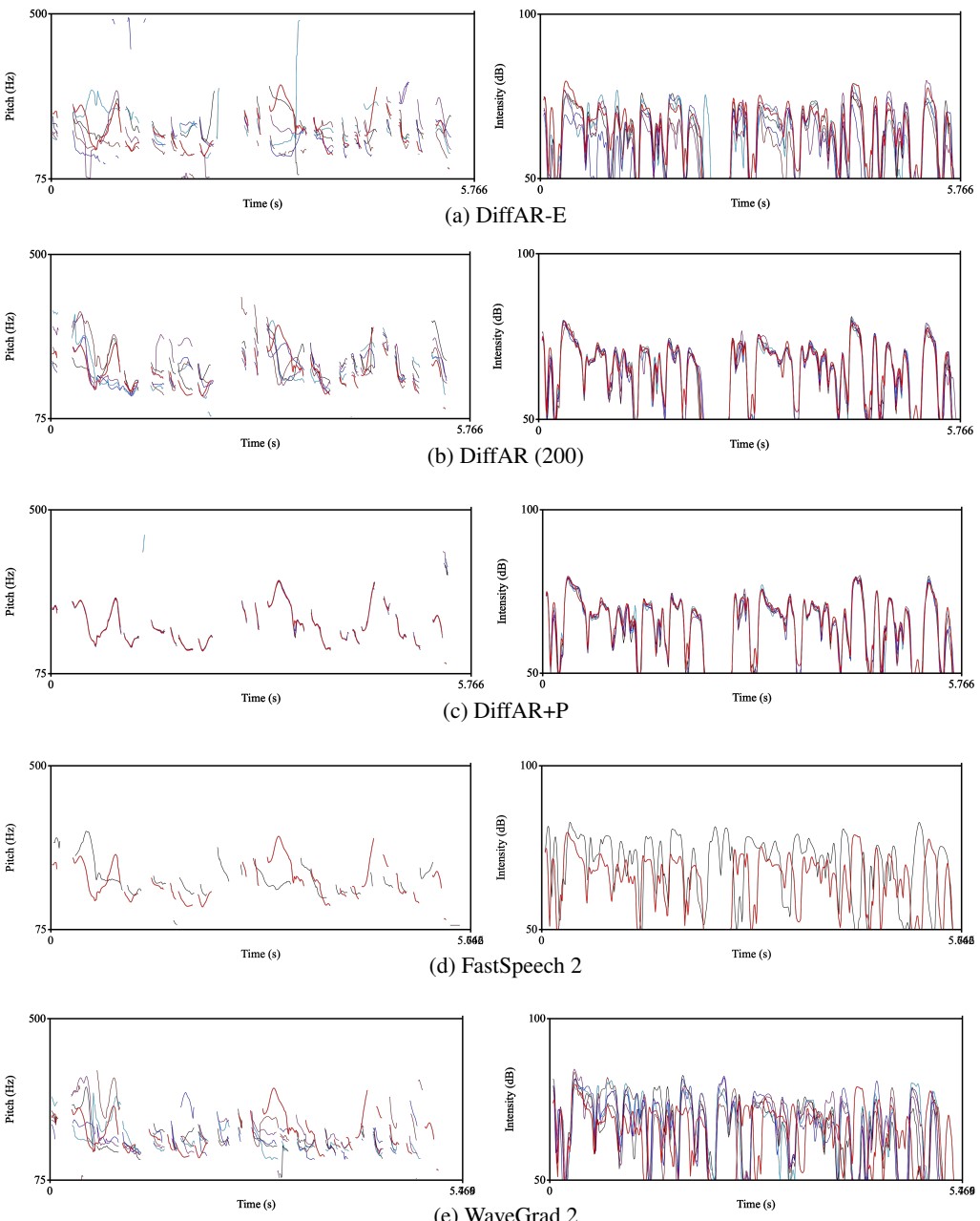

Figure 3: Comparing the energy and pitch of five samples that describe the same text, with the desired energy and pitch values marked in red.

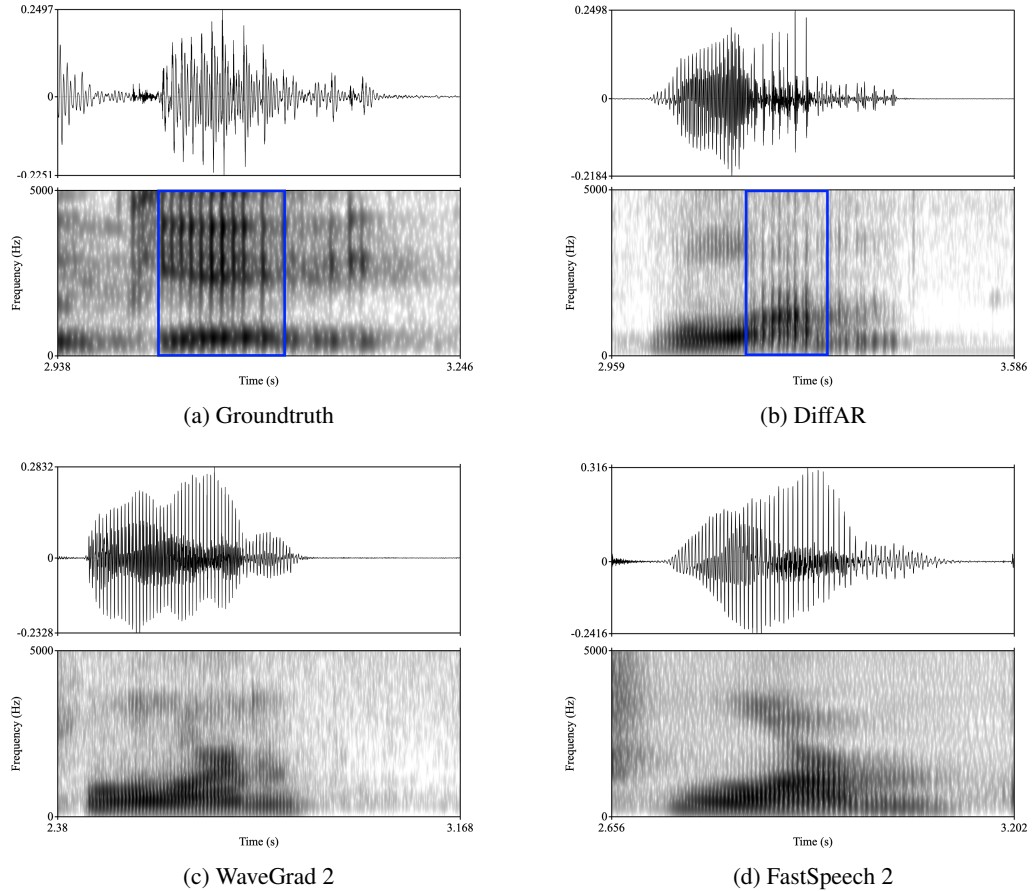

Figure 4: Displaying the vocal fry phenomenon across various models

signal. Since many models focus on estimating the signal's amplitude, often by deriving its spectrogram, this complexity poses an even greater obstacle, maybe preventing such models from faithfully reproducing the vocal fry phenomenon.

Figure 4d and Figure 4c shows generation of the same utterance with *FastSpeech 2* (former) and *WaveGrad 2* (latter). None of them generate vocal fry.

*DiffAR*, as an end-to-end model that generates the raw waveform, could capture the vocal fry phenomenon by integrating both the phase and amplitude components throughout the synthesis process. The result is a synthesis incorporating sound elements more closely resembling human speech, which likely contributes to the positive subjective results observed in our evaluations. An example can be found in Figure 4b, where the creaky area is highlighted in blue.

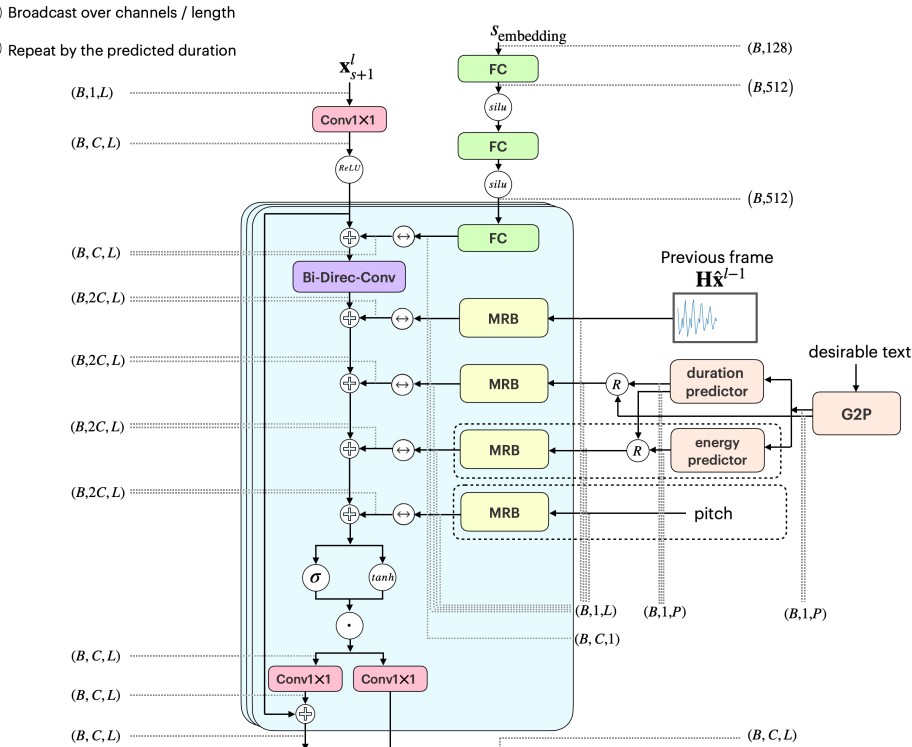

Figure 5: A detailed overview of a single residual layer

## C  DETAILED ARCHITECTURE

A detailed overview of a single residual layer is depicted in Figure 5, where $P$ represents the number of phonemes in the current frame, $L$ corresponds to the frame length, B indicates the batch size, which is set to 1 during inference, and C represents the number of residual channels, set to 256.

In our model, the duration and energy predictors are small neural networks trained and validated using the original LJ-Speech data partitioning. In both cases, the training objective was to minimize the Mean Squared Error (MSE) loss.

### C.1  DURATION PREDICTOR

The duration predictor takes a series of phonemes as input and predicts their expected durations. The network architecture consists of a phoneme-embedding layer ($|\mathcal{Y}| = 73, 128$), followed by a 1-D convolutional layer (128 input channels, 256 output channels, kernel size 5, stride 1, padding 2), a ReLU activation function, a normalization layer, a dropout layer with $p = 0.5$ dropout rate, and finally, a linear layer (256 input features, 1 output feature). During training, the timing was obtained using a phoneme alignment procedure (McAuliffe et al., 2017).

### C.2  ENERGY PREDICTOR

The energy predictor is a network that takes a series of phonemes as input and predicts their energy levels. The network architecture consists of a phoneme-embedding layer ($|\mathcal{Y}| = 73, 128$), followed by a sequence of two identical layers. Each of these layers consists of a 1-D convolutional layer (128 input channels, 256 output channels, kernel size 7, stride 1, padding 2), followed by another 1-D convolutional layer (128 input channels, 256 output channels, kernel size 5, stride 1, padding 2), a ReLU activation function, a normalization layer, and a dropout layer with $p = 0.5$ dropout rate. Finally, the second layer is followed by a linear layer (256 input features, 1 output feature). During training, the energy values for each phoneme were calculated as the square root of the average energy within each phoneme's duration.

# D COMPREHENSIVE COMPARISON TO OTHER METHODS

In addition to comparing *DiffAR* to *WaveGrad 2* and *FastSpeech 2*, we conducted a comprehensive comparison that includes both acoustic models (i.e., decoders) and *end-to-end* models. The models considered are *VITS* (Kim et al., 2021) , *Grad-TTS* (Popov et al., 2021), *Pro-DIFF* (Huang et al., 2022b), and *DiffGAN-TTS* (Liu et al., 2022). We conducted a MUSHRA test to evaluate audio quality[8] and examined factors such as the stochasticity of synthesis as all are one-to-many models, the architectural complexity of the models, and their ability to control stylistic features.

*VITS* is an *end-to-end* model that incorporates VAE (Kingma & Welling, 2013), Normalizing Flow (Rezende & Mohamed, 2015), MAS algorithm (Kim et al., 2020), and adversarial training for the TTS task. During training, it learns latent variables from linear spectrograms obtained from STFT, indicating that the synthesis doesn't directly operate on the waveform. Moreover, it includes a reconstruction loss involving the Mel-spectrogram representation and an adversarial loss on the output, which may be unstable during training.

In terms of qualitative metrics, we performed a MUSHRA test following the methodology described in Section 3. The results are presented in Table 4.

To assess the level of stochasticity in the model, we utilized the method outlined in Appendix A. The results in Figure 6a indicate a degree of stochasticity in the model, albeit to a limited extent. Notably, the pitch values in different draws exhibit a very similar pattern with slight shifts in the timeline, a behavior also observed in the energy values. A plausible explanation for this phenomenon is that the stochasticity in *VITS* primarily stems from the use of a stochastic time predictor, synthesizing speech at varying rates. However, it appears that it does not result in unique phenomena or prosody in the speech.

Each table is based on a different set of listeners hence the groud truth (GT) is not the same

Table 4: VITS (Kim et al., 2021)

| Method | ↑MUSHRA |
|---|---|
| Ground truth | $74.9 \pm 2.2$ |
| DiffAR (200 steps) | $69.1 \pm 2.2$ |
| DiffAR (1000 steps) | $\mathbf{71.5 \pm 2.2}$ |
| VITS | $69.0 \pm 2.3$ |

Table 5: Grad-TTS (Popov et al., 2021)

| Method | ↑MUSHRA |
|---|---|
| Ground truth | $73.7 \pm 2.4$ |
| DiffAR (200 steps) | $\mathbf{69.4 \pm 2.5}$ |
| DiffAR (1000 steps) | $67.7 \pm 2.6$ |
| Grad-TTS | $68.5 \pm 2.5$ |

Table 6: ProDiff (Huang et al., 2022b)

| Method | ↑MUSHRA |
|---|---|
| Ground truth | $70.0 \pm 2.1$ |
| DiffAR (200 steps) | $66.6 \pm 2.4$ |
| DiffAR (1000 steps) | $\mathbf{67.5 \pm 2.3}$ |
| ProDiff | $64.6 \pm 2.4$ |

Table 7: DiffGAN-TTS (Liu et al., 2022)

| Method | ↑MUSHRA |
|---|---|
| Ground truth | $71.2 \pm 2.0$ |
| DiffAR (200 steps) | $\mathbf{69.5 \pm 2.1}$ |
| DiffAR (1000 steps) | $68.4 \pm 2.2$ |
| DiffGAN-TTS | $68.0 \pm 2.2$ |

We turn now to the models *Grad-TTS*, *Pro-DIFF*, and *DiffGAN-TTS*, all fall into the category of diffusion-based acoustic models. These models, given text input, generate a spectrogram (and not work on the waveform directly, as we do), and subsequently, a *vocoder* is employed to produce the waveform. A common characteristic among these models is the desire to accelerate the diffusion process, often impacting audio quality, the stochasticity of synthesis, or the model's complexity.

*Grad-TTS* explicitly manages the trade-off between sound quality and inference speed. A significant modification involves initiating the diffusion process from noised acoustic information $\mathcal{N}(\mu, \sigma)$ rather than white noise $\mathcal{N}(0, I)$. This modification enables synthesis with a very limited number of

---

[8]Due to limited resources we chose MUSHRA over MOS as it is more robust and less subjective.

steps. *ProDiff* adopts the generator-based method and also incorporates knowledge distillation and the DDIM method to enable synthesis with only 2 diffusion steps. *DiffGAN-TTS* achieves a significant reduction in synthesis time by decreasing the number of diffusion steps, sometimes even to a single step. This is achieved through adversarial training of a GAN, which can occasionally introduce instability in the training process.

We conducted a MUSHRA test to assess the quality of *Grad-TTS* , *Pro-DIFF*, and *DiffGAN-TTS* compared to *DiffAR*. The models evaluated were *Grad-TTS* with $T = 1000$ diffusion steps, *Pro-DIFF*, and *DiffGAN-TTS* with $T = 4$ steps. The results are presented in Tables 5, 6 and 7. Based on the MUSHRA values, it is evident that our model produces speech characterized by higher quality compared to the evaluated models.

We explored the stochasticity of the models using the methodology outlined in Appendix A. Figures 6b 6d 6c depict the results. In all cases, the energy and pitch values were either identical or very similar across different samples. This suggests that, despite utilizing diffusion models, these models exhibit reduced or negligible stochasticity. While the smaller manifold of the spectrogram compared to that of the waveform might contribute to the decreased stochasticity, it may not be the sole factor influencing this phenomenon.

Regarding *DiffGAN-TTS* and *ProDiff*, both models significantly decrease the synthesis time by minimizing the number of diffusion steps, leading to an outcome that closely resembles a deterministic model. For *GradTTS*, initiating the diffusion process not with white noise and shortening the diffusion process diminishes the model's stochasticity. Figure 6b illustrates that using $T = 1000$ diffusion steps produces almost identical draws. Hence, despite the presence of numerous diffusion steps, the guidance is highly explicit, leaving minimal room for stochasticity.

In Figure 7 we see no real generation of vocal fry by any of these methods as seen by our model (cf. Figure B).

Despite the advantage of faster synthesis, changing and accelerating the diffusion process in all models decreased their stochasticity and creativity as one-to-many models. It appears that making the mapping between input (text) and output (speech) more deterministic also aims to reduce underfitting. *Grad-TTS* mentioned the possibility of an *end-to-end* model, but the results are not of high quality for a meaningful comparison.

Regarding *VALL-E* (Wang et al., 2023) and *Voicebox* (Le et al., 2023), both represent state-of-the-art models trained on large-scale datasets (more than ten thousand hours), hence will not be comapred here. While Voicebox as a *decoder* and VALL-E as an *end-to-end* model excels on large-scale datasets, replicating their success on LJspeech and VCTK proved challenging.

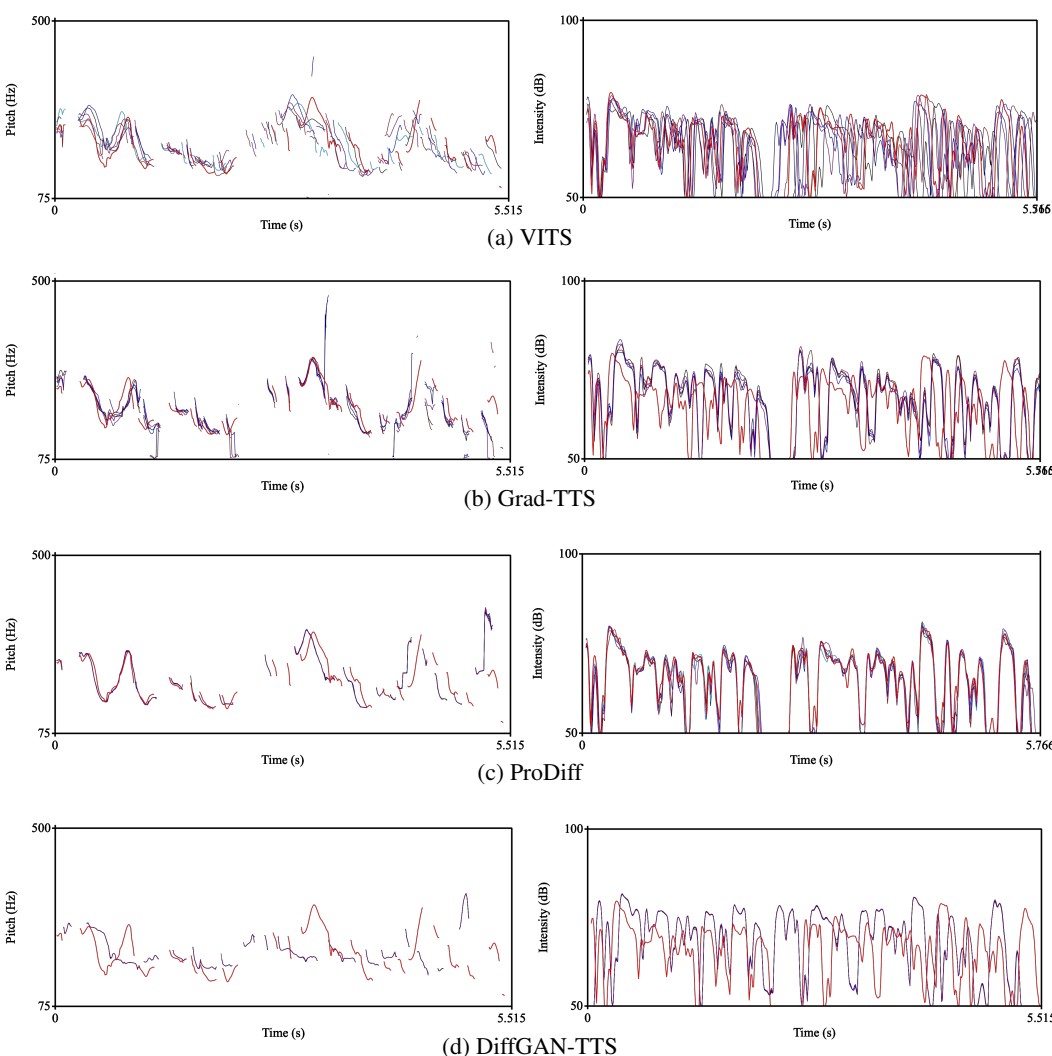

Figure 6: Comparing the energy and pitch of five samples that describe the same text, with the original energy and pitch values marked in red.

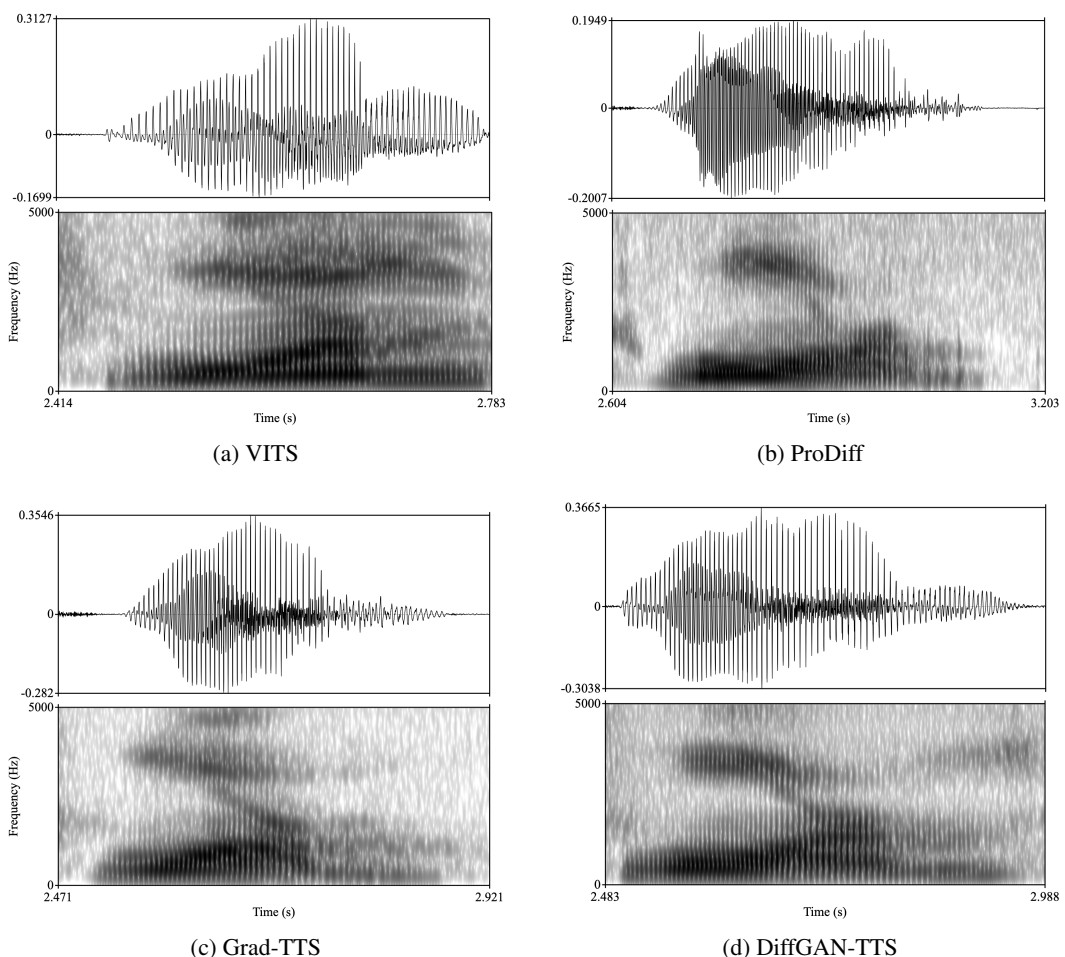

(a) VITS

(b) ProDiff

(c) Grad-TTS

(d) DiffGAN-TTS

Figure 7: Displaying the vocal fry phenomenon across various models

## E    COMPUTATIONAL LIMITATIONS AND SYNTHESIS TIME

Existing models face a notable challenge in training and synthesizing extremely long texts due to GPU computational constraints (Ren et al., 2020). However, with its autoregressive architecture, our model might handle this while preserving a consistent signal structure.

Figure 8 presents the analysis of the synthesis process of three models: *DiffAR (200)*, *WaveGrad 2*, and *FastSpeech 2*, where each time we doubled the number of words in the text and tested the maximum GPU consumption throughout the process. Each point on this graph was created by executing the corresponding model on GPU NVIDIA A40 with a memory of 48GB.

For the last two models, the GPU consumption escalates with an increase in text length up to a certain threshold where it hits a limit and triggers an out-of-memory error. For the *WaveGrad 2* model, this occurs post-processing 512 words; in the case of *FastSpeech 2*, it happens after 1024 words. Contrarily, our model maintains a consistent memory consumption level, an order of magnitude lower than the other models, offering controlled efficiency.

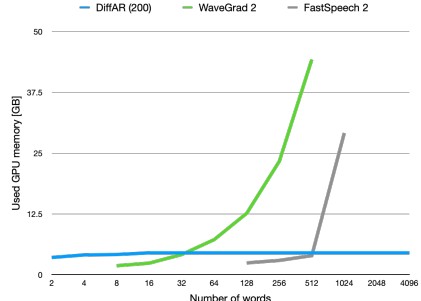

Figure 8: Used memory versus text length.

Table 8: Real-time Factor (RTF)

| Method | ↑RTF |
| --- | --- |
| Fastspeech2 | 2.79 |
| Wavegrad2 | 5.14 |
| VITS | 0.09 |
| ProDiff | 0.25 |
| GradTTS | 3.24 |
| DiffGAN | 0.06 |
| DiffAR | 31.06 |

A significant characteristic of the TTS task is the synthesis time. Recent efforts have aimed at achieving low RTF values, enabling fast synthesis for real-time and everyday applications. Huang et al. (2022b); Liu et al. (2022) Numerous models, particularly diffusion models, acknowledge the trade-off between audio quality and inference duration. By controlling the synthesis duration, they imply a decline in performance when the synthesis time is significantly reduced. (Popov et al., 2021; Jeong et al., 2021)

While achieving high-quality synthesis, a notable limitation of *DiffAR* is the extended synthesis time associated with the use of diffusion models and the inherent limitation in the autoregressive approach, which is sequential by definition. In Table 8, it is noticeable that DiffAR reaches lower RTF performance compared to the other models, and is not performed in real-time.

Another trade-off worth noting is between the RTF and the level of stochasticity in the generated signal. It can be seen in Figures 6 that all the models except *WaveGrad 2* generate almost the very same signal for every inference, while *DiffAR* generates a slightly new version at each inference call. Accelerating the diffusion process, providing an explicit guidance (i.e., Initializing the signal not with white noise (Popov et al., 2021)), and incorporating deterministic components - all harm the model's ability to generate a new version of the waveform and be utilized as a one-to-many model. This, in turn, also influences the generation of the prosodical features of the signal (such as vocal fry).

There are numerous strategies to expedite the synthesis process while still maintaining the autoregressive nature: Reducing the number of steps in the diffusion process (e.g., using DDIM Song et al. (2020), which involves trade-offs as previously discussed) or even by developing a parallelized algorithm (Oord et al., 2018). The focus of our work was to generate a realistic signal with prosodical features and with natural variability. Hence, addressing this issue will be deferred to future work.

## F    EXTENSION TO MULTIPLE SPEAKERS

While the traditional TTS task typically involves a single-speaker dataset, other research directions include using multiple speakers or languages and incorporating emotional characteristics or background noises guided by text. Additionally, combining multiple speakers in a single text is another potential avenue.

Various methods exist for performing these tasks, particularly in the context of diffusion models. The main approaches include explicit conditioning (Ho & Salimans, 2022) or utilizing external guidance during the update of the diffusion procedure (Dhariwal & Nichol, 2021). For DiffAR, we decided to investigate working in the multi-speaker scenario. Our approach involves leveraging the model's versatility by incorporating the speaker's embedding into the synthesis process. We explored this option using *Titanet* embedding (Koluguri et al., 2022) and VCTK dataset (Veaux et al., 2017).

Figure 9 illustrates the architectural modification we implemented, where $v_{embedding}$ represents the speaker embedding obtained from the *Titanet* network output.

Examples of multi-speaker generation can be found in our model's GitHub repository[9].

---

[9]https://github.com/RBenita/DIFFAR

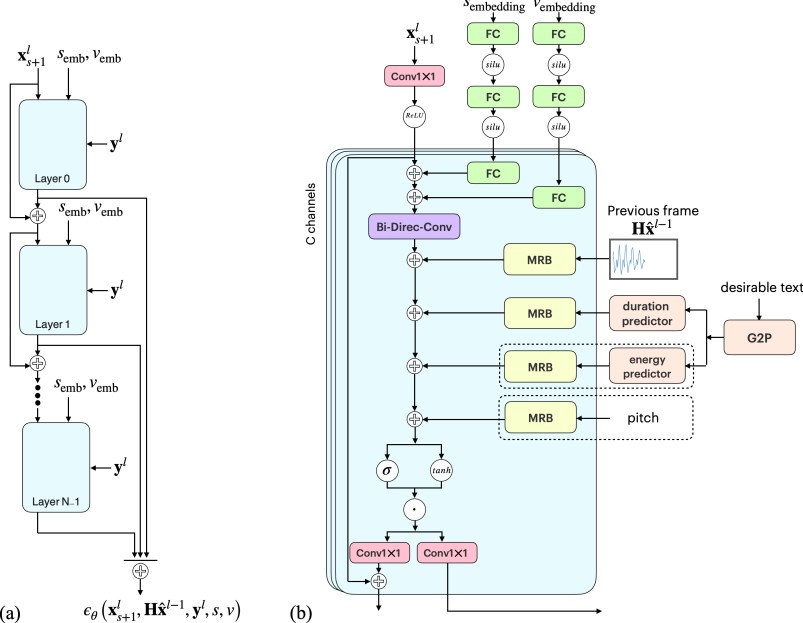

Figure 9: (a) A general overview of the structure of the residual layers and their interconnections. (b) A detailed overview of a single residual layer within the multi-speaker scenario.

Another potential approach would involve utilizing the autoregressive manner for in-context learning. Given an initial frame in a specific voice and a desired text, the model would continue the speech in the same given style without relying on additional information. However, addressing this issue goes beyond the scope of this paper.

