# OpenReview forum: "DiffAR: Denoising Diffusion Autoregressive Model for Raw Speech Waveform Generation"
_ICLR.cc/2024/Conference — ICLR 2024 poster_

### Official Review · Reviewer_u4XD · 2023-10-29

**Soundness:** 3 good
**Presentation:** 3 good
**Contribution:** 2 fair
**Rating:** 5
**Confidence:** 4

**Summary:**

The paper proposes an auto-regressive diffusion TTS framework that can continuously generate long sentences with almost constant GPU memory cost. The model generates more natural speech, especially details like vocal fry, compared to baseline models. System comparison experiments and ablation study on the conditional information are conducted.

**Strengths:**

1. The proposed method can generate some voice details including vocal fry, which is more natural than the baseline models.
2. The paper is easy to read.

**Weaknesses:**

1. Compared with FastSpeech2 and WaveGrad2, the proposed method consumes even more GPU memory in normal-length sentences.
2. Lack of comparison with simple baselines. See questions.

**Questions:**

1. More details about the linguistic representation need to be clarified. In the conditional setting, is the linguistic representation ${y}^l$ aware of the whole sentence text information or just aware of the text information of the corresponding frame? Is there a length regulator to align the text representation to the sample points?
2. What is the practical scenario of synthesizing extremely long sentences (>512 words)? Such long sentences are rare in natural language. Compared with FastSpeech2 and WaveGrad2, the proposed method consumes even more GPU memory in normal-length sentences, which is below 512 or 32 words. There are obvious baseline methods to generate long sentences, such as splitting sentences or linguistic representation into chunks, synthesizing them separately, then concatenating the waveform. Under this setting, both of FastSpeech2 and WaveGrad2 can generate unlimited waveform durations while preserving the computational resources, and with even lower computation. Could the proposed method outperform these baselines by scaling down the computation cost to match these models?
3. Although there is a justification of the choice of a long frame length in the unconditional setting, the author did not explain why they also used such a long frame in the conditional setting. There is also no ablation study on this. Note that these two settings are different in aspect of the global information. The global information provided by the long frame length in the unconditional setting can be provided by linguistic representation ${y}^l$ in the conditional setting. If the frame length can be shortened without performance drop in the conditional setting, the GPU memory could be reduced further, but the proposed method would be like a vocoder rather than a TTS system.

---

> ### Author Response · Authors · 2023-11-20
> **Response to reviewer u4XD**
>
> Thank you for the comments. We would like to address each of them individually in a comprehensive manner.
>
> **Q1:**
>
> Our model generates time and energy predictions by incorporating information from the entire text. One of the advantages of this approach is its ability to facilitate more stable conditioning frameworks. If the prediction were local, the entire speech would likely be less smooth or stable.
> Regarding the length regulator, we consistently utilize $\alpha=1$ (normal speed) and extend the signal based on the predictions of the duration predictor to ensure alignment with the actual samples. We will provide further refinement and will change the model architecture figure accordingly.
>
>
> **Q2:**
>
> One of the benefits of the diffusive method and working directly on the waveform is the inherent stochasticity of the signal. Therefore, if we partition the signal into independent segments and then concatenate them, there's a possibility that one segment may be spoken in a certain prosody, while another segment may be spoken differently. In such cases, even if the concatenation is of excellent quality, the delivery of words may not be continuous in terms of prosody. Hence, the significance of autoregressive processes and the transfer of information over time. For models that lack stochasticity, the suggested method of concatenation may be suitable, but it comes with the drawback of determinism.
>
> While very long signals like 512 words are indeed uncommon in everyday language usage (except in cases of reading an article or a book), it is noticeable that for an E2E model like Wavegrad2, the transition occurred even for texts of reasonable length (32 words). One possible strategy to decrease GPU consumption is to **simplify or substitute the denoiser architecture.** As an illustration, we executed a model with a lower number of residual channels (C=128). This resulted in a reduction in GPU consumption, although not to a sufficient extent. (A figure for the analysis is available on our Git repository) We haven't had the opportunity to conduct a comprehensive comparison regarding quality results, but initial observations suggest promising and good-sounding outcomes. Reducing the frame length could be considered as another method for decreasing GPU consumption, but we will address this in the response to question 3.
>
> We recognize the significance of optimizing the synthesis process, and it presents a direction for further research. Another potential research direction involves applying the autoregressive method to other time-dependent domains and assessing its effectiveness. For instance, in video synthesis, it's likely that GPU consumption over time may increase more significantly. Alternatively, the synthesis of DNA chains serves as an example for very long signal lengths. Exploring the application in these contexts presents interesting avenues for further research.
>
> **Q3:**
>
> Indeed, for the unconditional case, choosing a long frame is more comprehensible since generating a continuous signal requires obtaining all the necessary information directly from the waveform. In the conditional model, choosing the frame length involved several considerations:
> - Balancing between memory consumption and synthesis time: If the frame lengths were very small, the GPU consumption at any given time would indeed be lower. However, simultaneously, more serial rounds of diffusion would be required to generate a given text. This would result in a longer synthesis time.
> - While the representation $y^l$ from the past is valuable, from our perspective, it cannot substitute conditioning on past waveform values. This limitation arises from the stochasticity inherent in diffusion models and the waveform - given a certain representation $y^l$, it can be heard in many prosodies. Hence, to achieve natural-sounding continuity, referencing the past waveform is essential.
> - In certain cases, it might be acceptable to rely on a relatively recent history. However, for long phonemes or silent phonemes (which can extend up to 200 ms or more), ensuring continuity requires information from a more distant history of the waveform signal.
> - During the model's development, we acknowledged the significance of processing complete phonemes. Consequently, when conditioning synthesis on phoneme segments (leading to shorter conditioning), the model may struggle to discern the relative portion of the resulting phoneme. Our observations indicate that working with full phonemes produces more robust and "healthier" results.

---

### Official Review · Reviewer_9k2u · 2023-10-31

**Soundness:** 3 good
**Presentation:** 2 fair
**Contribution:** 2 fair
**Rating:** 6
**Confidence:** 4

**Summary:**

This proposes a novel form of generative models, which modifies the prominent denoising diffusion probabilistic models into an autoregressive model for end-to-end speech synthesis. The main idea in DiffAR is to sequentially condition each frame on a portion of the previously generated one for diffusion noise prediction. An extensive performance analysis is presented in the paper, demonstrating high-fidelity synthesis and temporal coherence.

**Strengths:**

The strengths of this work are summarized in the following:
1. Novelty: The proposed DiffAR is novel and technically sound. This modeling approach is fundamentally superior over other non-autoregressive diffusion models with its support of unlimited speech duration.
2. Reproducibility: The authors also provide the codes and samples on anonymous github, which make the proposed model transparent and reproducible.

**Weaknesses:**

There are two fatal issues in this paper:

1. Weak baselines and limited reference: The author should acknowledge and compare their proposed model against other works in a relevant research direction (i.e., diffusion-based TTS), e.g., FastDiff, Diff-TTS, Grad-TTS, PriorGrad, ProDiff, etc. It is understandable that comparing with all the conventional methods is computationally prohibitive, yet the authors should at least discuss the pros and cons of these methods in comparison to DiffAR. The latest diffusion-based TTS model that the authors referred to was WaveGrad 2 (released in 2021), which is not up-to-date and not the SOTA in this research direction. I am skeptical whether this baseline is strong enough, and I suggest the authors to use a more recent TTS model as for a stronger baseline.

2. Implicit description on synthesis time: In section 5.4, while the section is titled "COMPUTATIONAL LIMITATIONS AND SYNTHESIS TIME", I expect to see exact numbers or a graph reporting the inference speed of DiffAR. However, the authors only show the memory usage of DiffAR and implicitly describe the limitation of DiffAR in synthesis time, which makes it unclear how much we need to sacrifice in efficiency to exchange for an improved quality. It would be unfair if we focus on quality and compare DiffAR with FastSpeech 2, since FS2 is clearly much more efficient in time. And, conceivably, as an autoregressive model similar to WaveNet, DiffAR should be quite sensitive to the length of the waveform to be generated.

**Questions:**

My questions are stated in the weakness above.

---

> ### Author Response · Authors · 2023-11-20
> **Response to reviewer 9k2u (part 1/2)**
>
> Thank you for the comments. You brought up interesting points, and we would like to respond to them comprehensively.
>
> **TL;DR:**  The models **FastDIFF, DiffTTS, Grad-TTS, PriorGrad, and ProDiff**  utilize diffusion-based approaches, functioning either as acoustic models generating a mel-spectrogram (thus requiring a vocoder for waveform synthesis) or as diffusion-based vocoders. A common feature among most of these models is the aim to accelerate the diffusion process, often at the expense of audio quality, the stochasticity of synthesis, or the simplicity of the model.
>
> **Q1:**
>
> In hindsight, we acknowledge that our comparison to existing models may not have been sufficiently comprehensive. Considering your feedback,  we added a comparison and explanation here and in the paper for the models you mentioned and others.
>
> Our goal was to develop an E2E model that, in addition to being constructed as a single unit, conducts the synthesis process directly on the waveform. We utilized qualitative metrics for comparison, such as MOS and MUSHRA, but also delved into examining the stochasticity in the model (as a "one-to-many" model) and the controllability of stylistic features. In our responses to question 2, we will also address the synthesis time comparison.
>
> **DiffTTS** is among the pioneering models that integrate the Text-to-Speech (TTS) task with diffusion. It utilizes the DDIM sampling equation to enable faster inference speed for Mel-spectrogram generation. **To the best of our knowledge, there is no publicly available implementation of DiffTTS, but based on official samples in the Git repository,** it seems to produce good-sounding results. However, the use of a vocoder occasionally introduces metallic or noisy sounds.
>
> **FastDIFF** incorporates noise scheduling algorithms and predicts faster sampling noise scheduling than the one used in training. The model is mainly employed as a vocoder component, but we recognize the potential for **FastDIFF-TTS** as an end-to-end model. **Given its official comparison in the paper and the Git repository, we decided to perform a comparison with ProDiff, a model written by joint authors and published later.**
>
> **PriorGrad** serves both as a vocoder and as an acoustic model. It formulates the sampling and training procedures to utilize an adaptive prior derived from data statistics. As an acoustic model, PriorGrad achieves slightly inferior results compared to GradTTS with T=10 diffusion steps. **Consequently, we chose to compare our model's results to the Grad-TTS with T=1000 diffusion steps.**
>
> **Grad-TTS** explicitly manages the trade-off between sound quality and inference speed. A notable modification involves initiating the diffusion process from $N(\mu, \sigma)$ rather than white noise.
> **ProDiff** adopts the generator-based method and also incorporates knowledge distillation and the DDIM method to enable synthesis with only 2 diffusion steps.
>
> To compare the quality of **Grad-TTS** and **ProDiff** , we conducted a MUSHRA test following the methodology described in the paper. **The models evaluated were Grad-TTS with T=100 diffusion steps and with ProDiff T=4.**
>
> | Model      | MUSHRA |
> | ----------- | ----------- |
> | GT     | 70.0 &plusmn; 2.1     |
> | DiffAR 200   | 66.6 &plusmn; 2.4      |
> | DiffAR 1000   | **67.5 &plusmn; 2.3**     |
> | ProDiff  | 64.4 &plusmn; 2.4      |
>
>
> | Model      | MUSHRA |
> | ----------- | ----------- |
> | GT     | 73.7 &plusmn; 2.4       |
> | DiffAR 200   | **69.4 &plusmn; 2.5**        |
> | DiffAR 1000   | 67.7 &plusmn; 2.6      |
> | Grad-TTS   | 68.5 &plusmn; 2.5      |
>
>
> (Each table is based on a different set of listeners, hence GT is not the same)
>
> Despite the advantage of fast synthesis, changing and shortening the diffusion process in both models led to a **decrease in their stochasticity and creativity as one-to-many models.** Regarding ProDiff, different syntheses for the same text were nearly identical in terms of pitch and energy values.
> In the case of GradTTS, initiating the diffusion process not with white noise and significantly shortening the diffusion process diminishes the stochasticity of the model. In the example provided on the website, we can see that **using T=1000 steps produces almost identical draws. Therefore, even with numerous diffusion steps, the guidance is highly explicit, leaving minimal room for stochasticity.** Furthermore, it is not possible to directly control characteristics throughout the synthesis; instead, changes can only be made indirectly by adjusting phoneme duration.
>
> **Grad-TTS mentions the option of an E2E model, but the results are not of high quality for meaningful comparison.**

---

> ### Author Response · Authors · 2023-11-20
> **Response to reviewer 9k2u (part 2/2)**
>
> **Q2:**
>
> In retrospect, we acknowledge that more explanation could have been given Regarding the synthesis time. In response to your question, we conducted the **RTF values** for the models we compare so far.
>
>
> | Syntax      | RTF |
> | ----------- | ----------- |
> | Fastspeech2     | 2.79     |
> | ProDiff  | 0.25     |
> | DiffGAN  | 0.06      |
> | GradTTS  | 3.24      |
> | VITS  | 0.09     |
> | Wavegrad2   | 5.14       |
> | DiffAR  | 31.06      |
>
>
> As mentioned earlier, there is a significant tradeoff, particularly in diffusion models, between synthesis time, synthesis quality, and the level of creativity or stochasticity. The primary focus of our paper was to introduce a novel approach which utilizes the information in the waveform (phase and amplitude). We aimed to develop a model capable of generating natural speech phenomena and facilitates stochasticity as a one-to-many model.
> Following your comment, we acknowledge a potential research direction, alongside exploring architectural changes, involves maintaining synthesis quality and stochasticity while reducing synthesis time.

---

### Official Review · Reviewer_iPTb · 2023-11-01

**Soundness:** 3 good
**Presentation:** 3 good
**Contribution:** 2 fair
**Rating:** 6
**Confidence:** 3

**Summary:**

This paper describes an audio waveform synthesizer based on a diffusion process that can be conditioned on previous overlapping output frames so can be used in an autoregressive manner. The diffusion model itself only captures dependencies between overlapping frames and not between distant frames except through conditioning variables (e.g., speech synthesis parameter trajectories) that might stretch across longer timescales. The audio instantiations are assumed to be conditionally independent given this information. In the experiments, these frames are 500ms long and overlap by 250ms with each other.

The system is evaluated on TTS for the LJ Speech single-talker corpus. It uses separate models for predicting duration, energy, and pitch, and pronunciation of a given transcript, then this diffusion model as vocoder to synthesize it. Listening tests with 45 raters rate the proposed method as 3.9 percentage points higher than WaveGrad 2 and 2.0 percentage points higher than FastSpeech 2 improved in a MUSHRA task. Putting the synthesized speech through a Whisper ASR yields lower CER and WER on the proposed method than WaveGrad2, while the same comparison shows slightly lower CER and WER for FastSpeech 2 improved than the proposed approach.

**Strengths:**

* Paper is well written and well organized
* Excellent literature review
* Good derivation and mathematical exposition of the method
* Listening tests seem well conducted, clear task, well executed.

**Weaknesses:**

One of the main benefits of diffusion models is that they can accurately model all of the samples in a waveform jointly. This paper proposes breaking this into a series of conditionally independent frames to allow them to be created in order, to expedite generation, but does not acknowledge this strong change in assumptions from full joint modeling to conditional independence. Given this change in assumptions, it is not clear to me what the difference is between this approach and other diffusion-based vocoders. This difference should be emphasized and elaborated in the paper.

The comparison with other methods does not seem able to distinguish between differences in the TTS front end and back end between the methods. The different methods differ in both, so it is not clear if differences in performance can be attributes to the proposed method, which is really only concerned with the backend vocoder and not the system for generating the conditioning information. It is also not clear why the particular TTS front end was selected, as it seems less sophisticated than tacotron2, for example, which has been around for several years.

**Questions:**

What is the difference between this method and other diffusion-based vocoders?

---

> ### Author Response · Authors · 2023-11-20
> **Response to reviewer iPTb**
>
> Thank you for your detailed and constructive feedback. You provided insightful comments regarding the model architecture, and we would like to respond to them.
>
> Our main goal was to develop an E2E model that would be able to work directly on the waveform and produce high-quality speech. When dealing with relatively long texts (10 sec or more) , training or synthesizing the waveform jointly, especially in diffusion models, becomes almost impossible mainly due to computational and memory constraints. In order to do so, we divided the signal into frames (250 msec). We assumed conditional independence (Markovian) between frames in Eq (3) and we will state that in the paper. This is a common assumption in many speech recognition and speech generation models, and empirically produces very high quality speech.
>
> Notably, diffusion models like Wavegrad2 and GradTTS also address this challenge by conducting the learning process on segmented portions of the signal, rather than training the model on the entire text in a holistic manner. In other words, their learning process is also local with a relatively limited receptive field, tackling the same computational difficulty.
>
> An alternative solution (not ours) would have been training an acoustic model (such as Fastspeech2, DiffTTS, GradTTS, etc.) given a text, it generates linguistic characteristics with a lower dimension of the waveform.
>
> The distinctiveness of our model, compared to acoustic models, lies in its choice to skip intermediary linguistic features such as a spectrogram and to forgo the use of a vocoder. The notion is that working directly on the waveform enables the generation of more natural speech. It is important to note that both the duration predictor and the energy predictor are simple networks (comprising few fully connected layers). They cannot be considered decoders on their own. (For instance, in the case of Fastspeech2 as an acoustic model, similar predictors are also used, but they do not independently extract linguistic features from the text). Consequently, DiffAR should not be regarded as a vocoder or as a concatenation of an acoustic model and a vocoder. Therefore, we opted for a qualitative comparison with **Fastspeech2** as an acoustic model and with **Wavegrad2** as an end-to-end model.
>
> As for **Tacotron2**- the model functions as an acoustic model. It converts text into a Mel-spectrogram and uses Wavenet as a vocoder for waveform generation. Operating deterministically, it consistently produces a specific signal, for a given text. In qualitative metrics, Fastspeech2, VITS, and GradTTS have shown higher Mean Opinion Score (MOS) values than Tacotron2.  Given that our model achieved better MUSHRA / MOS results than these models, we infer that in terms of audio quality, we outperform Tacotron2 as well.

---

### Official Review · Reviewer_a3Eo · 2023-11-01

**Soundness:** 4 excellent
**Presentation:** 2 fair
**Contribution:** 2 fair
**Rating:** 6
**Confidence:** 4

**Summary:**

This papers fills in a notable missing piece in the matrix of state of the art TTS methodologies - namely probabilistic diffusion based end to end TTS using an AR model. It does not rely on potentially limited intermediate representations (adopted by acoustic models) such MFCC, Mel-spectrograms and directly generates the raw waveform using overlapping frames. The author's claim that the data driven intermediate representation more effectively captures prosody (specifically, *vocal fry* in the paper) due to the presence of both amplitude and phase information. It offers an architecture that enables explicit conditioning of desirable information (energy, pitch and duration)

**Strengths:**

The key strength of the paper lies in providing a method for high quality speech synthesis, which can be conditioned not only on the input text but also vocal features such as energy and pitch.

The following points summarize the main strengths of the paper
- Arbitrary length waveforms can be generated because of an AR approach, both with respect to modelling but also computationally since memory consumption does not scale with sequence length (unlike the NAR approaches compared with in the paper)
- Experimental results include MUSHRA as a metric which is more robust/less subjective compared to MOS, outperforming sensible baselines
- Comprehensive ablation of different model components, which demonstrates their efficacy
- It allows variation of stylistic (prosody) features both by explicit conditioning and the inherent stochasticity of the proposed model (vs practically deterministic approaches like FastSpeech 2)

Overall, this paper represents a good, sound contribution to the TTS literature providing a method to synthesize high fidelity sound using a combination of theoretical and practical (overlapping frames) techniques.

**Weaknesses:**

The primary weakness of the paper lies in the somewhat limited experimental evaluation (with respect to dataset size/variety), and lack of references/comparisons to recent relevant literature.

The papers fails to adequately analyse closely related variants which are cited but not evaluated against - specifically DiffTTS (Jeong et al., 2021) and GradTTS (Popov et al., 2021). Due to the similarity in methodology, it is important to differentiate DiffAR where the only justification provided is the potentially better representation compared to using MelSpectrograms (which contain only amplitude information). However, capturing vocal fry without more comprehensive experiments (the work only compares against 2 model classes - FastSpeech2 + HiFi GAN and Wavegrad 2) is not enough to establish superiority of the proposed technique over variants which use acoustic features.


Moreover, an entire class of models which have established SOTA are not mentioned (VITS -> YourTTS -> VALL-E -> Voicebox (recent, past the ICLR cutoff)). All of these models outperform baselines mentioned in the current paper, and therefore the paper fails to establish its position and tradeoffs with respect to recent literature.

The overall theme is that due to either omission or lack of comparison (quantiative/qualitative), the paper feels like an alternative methodology that isn't convincing enough in its evaluation about outperforming/matching SOTA (which it does claim to do so). This is especially critical because of the key choice of choosing an *AR model* which are substantially slower than their non auto-regressive counterparts. Consequently for one to inherit the caveat of slow inference (RTF not measured in the paper), the model must have high quality and/or other advantages in order to justify the tradeoff.

### Major
- Single speaker dataset, with limited samples vs larger, multi speaker datasets (LibriTTS (Kahnetal et al, 2020), LibriSpeech, VCTK(Veauxetal et al,2016), LibriTTS(Zenetal et al, 2019))
- Quantitative/qualitative comparisons are provided against models (Fastspeech2 + HiFi GAN, Wavegrad 2) which are no longer SOTA (even as of Jan 2023) - missing important relevant references such as VITS, YourTTS, VALL-E.
> I acknowledge the difficulty of obtaining implmentations which is alluded to in the footnote mentioning that implementations of fastspeech 2s (Ren et al., 2020), Wave-Tacotron (Weiss et al., 2021) could not be obtained.
- Authors claim "Although there is a plethora of TTS systems available, our objective was to benchmark against the most high-performing and relevant models WaveGrad 2 and FastSpeech 2." without defining clearly how relevance is established since one can argue based on alternative methodologies and/or performance DiffTTS, GradTTS, DiffGAN-TTS, VITS, YourTTS are also extremely viable candidates. Even a qualitative, written explanation of the critical tradeoffs between the current work and the aforementioned models would help one understand the contribution of the work significantly better.

### Minor
- No benchmarks on synthesis speed (Real time factor), but based on the chosen schemes (DDPM + AR) it seems evident that the model will be quite slow
- No discussion on extending the current model to multi speaker and other prosodies (it is mentioned in the conclusion as future work)

I vote to reject the paper due to the aforementioned reasons, but I am willing to be corrected and change my score accordingly in case I misunderstood any aspect of the work and/or the literature.

**Questions:**

- What were the criteria for choosing baselines which are just said to be relevant? Why were the other E2E models (VITS, YourTTS, EATS) or Acoustic models (GlowTTS, DiffTTS, GradTTS) not compared against?
- How stable is the training of the model? Do you backpropagate into the energy and duration predictors (I do not think so based on the model)?
- How is pitch predicted at inference time?
- What was the proportion of intelligible vs random words in unconditional generation?

---

> ### Author Response · Authors · 2023-11-20
> **Response to reviewer a3Eo (part 1/4)**
>
> Thank you for the detailed comments. You raised interesting points, and we would like to respond to them comprehensively. In hindsight, we acknowledge that our comparison to existing models may not have been sufficiently comprehensive. Considering your feedback, we added a comparison and explanation here and in the paper for the models you mentioned and others. Furthermore, we answer here and in the paper the other concerns you raised.
>
> **TL;DR:** The models DiffTTS, GradTTS, and DiffGAN-TTS generate a spectrogram, and subsequently, a vocoder is employed to produce the waveform, while our model is a single end-to-end model that directly generates a waveform.
> Below (and soon in the paper), we compare our model to DiffGAN, GradTTS, and VITS. The models DiffTTS, VALL-E, Voicebox, and EATS have no implementation, while YourTTS architecture is very similar to VITS and focuses on slightly different tasks than ours, namely Zero-shot-multi-speaker. Details below.
>
> **Major:**
>
> As mentioned, our objective was to introduce an E2E model that, beyond being constructed as a single unit, conducts the synthesis process **directly on the waveform ​**​(without replying on an intermediate representations such MFCC, Mel-spectrograms), hence can captures prosody (such as vocal fry). In the comparison, we referred to qualitative metrics such as MOS and MUSHRA, but **it was also important for us to examine the stochasticity/creativity** (as a "one-to-many" model) as well as the controllability over stylistic features.
>
> The models DiffTTS, GradTTS, and DiffGAN-TTS all fall into the category of diffusion-based acoustic models. These models, given text input, generate a spectrogram, and subsequently, a vocoder is employed to produce the waveform. A common characteristic among most of these models is the desire to accelerate the diffusion process, often impacting audio quality, the stochasticity of synthesis, or the model's complexity.
>
> **DiffTTS** is among the pioneering models that integrate the Text-to-Speech (TTS) task with diffusion. It utilizes the DDIM sampling equation to enable faster inference speed for Mel-spectrogram generation. **To the best of our knowledge, there is no publicly available implementation of DiffTTS, but based on official samples in the Git repository, it seems to produce good-sounding results.** However, the use of a vocoder occasionally introduces metallic or noisy sounds.
>
> **Grad-TTS** explicitly manages the trade-off between sound quality and inference speed. A notable modification involves initiating the diffusion process from $N(\mu, \sigma)$ rather than white noise.
> **DiffGAN-TTS** achieves a significant reduction in synthesis time by employing adversarial training of a GAN, which can occasionally introduce instability in the training process.
>
> To compare the quality of **GradTTS** and **DiffGAN-TTS,** we conducted a MUSHRA test following the methodology described in the paper. **The models evaluated were GradTTS with T=1000 diffusion steps and DiffGAN-TTS with T=4.**
>
>
> | Model      | MUSHRA |
> | ----------- | ----------- |
> | GT     | 73.7 &plusmn; 2.4       |
> | DiffAR 200   | **69.4 &plusmn; 2.5**        |
> | DiffAR 1000   | 67.7 &plusmn; 2.6      |
> | GRAD-TTS   | 68.5 &plusmn; 2.5      |
>
>
> | Model      | MUSHRA |
> | ----------- | ----------- |
> | GT     | 71.2 &plusmn; 2      |
> | DiffAR 200   | **69.5 &plusmn; 2.1**       |
> | DiffAR 1000   | 68.4 &plusmn; 2.2      |
> | DiffGAN-TTS   | 68.0 &plusmn; 2.2      |
>
> (Each table is based on a different set of listeners, hence GT is not the same)
>
> **We also investigated the stochasticity of Grad-TTS and DiffGAN-TTS** by generating five syntheses of the same text. The charts illustrating these comparisons are available on the website and in the article. **In both cases, the energy and pitch values were found to be the same or nearly identical for different samples.** This indicates that, despite being diffusion models, the stochasticity in these models is reduced or non-existent. While the small manifold of the spectrogram compared to that of the waveform might contribute to the reduced stochasticity, it may not be the sole factor influencing the phenomenon.
>
> In the case of GradTTS, initiating the diffusion process not with white noise and significantly shortening the diffusion process diminishes the stochasticity of the model. In the example provided on the website, we can see that using **T=1000 steps produces almost identical draws. Therefore, even with numerous diffusion steps, the guidance is highly explicit, leaving minimal room for stochasticity.**
>
> DiffGAN-TTS drastically reduces synthesis time by minimizing the number of diffusion steps, **resulting in an outcome that closely resembles a deterministic model.**
>
> For the models DiffTTS, GradTTS and DiffGAN-TTS, explicit control over signal properties. While in some cases, phoneme duration can be indirectly adjusted, it remains the extent of controllability.

---

> ### Author Response · Authors · 2023-11-20
> **Response to reviewer a3Eo (part 2/4))**
>
> We turn now to the models - **EATS, VITS, YourTTS, VALL-E and Voicebox**, we would like to conduct a similar comparison.
>
> EATS, although a pioneer in E2E models, exhibits performance below the baselines. Moreover, it has a relatively complex architecture and is challenging to reproduce. Due to the absence of a public implementation, we decided to focus on the other models.
>
> **VITS** and **YourTTS** share a similar E2E architecture utilizing VAE, Normalizing Flow, MAS algorithm, and adversarial training for the TTS task. While VITS addresses the TTS scenario, YourTTS focuses on the Zero-One-shot task with multiple speakers and languages. Both models employ a stochastic time predictor to synthesize speech with diverse rhythms. However, for the same text the samples exhibit energy and pitch with very similar but shifted frames, as evidenced by examples attached to the paper and the Git repository. We should note that in the training of VITS and YourTTS, the latent variables z are based on linear spectrograms obtained by stft, so the synthesis doesn't directly work on the waveform. The reconstruction loss also involves the spectrogram, and the adversarial loss on the output may reduce stability during learning.
>
> **In terms of qualitative metrics,** we compared VITS as it is more closely related to the examined task. We will elaborate on the extension to multiple speakers in our responses to the questions.
>
> | Syntax      | MUSHRA |
> | ----------- | ----------- |
> | GT     | 74.9 &plusmn; 2.2     |
> | DiffAR 200   | 69.1 &plusmn; 2.2       |
> | DiffAR 1000   | **71.5 &plusmn; 2.2**      |
> | VITS  | 69.0 &plusmn; 2.3      |
>
>
> **VALL-E** and **Voicebox** represent state-of-the-art models trained on large-scale datasets. Voicebox can be utilized as a zero-shot TTS model by infilling missing parts in the Mel-spectrogram and as an acoustic TTS model. VALL-E is an E2E model that uses a Codec code, working indirectly on the waveform, to generate speech and handle the zero-one-shot task.
>
> While these models excel on large-scale datasets, replicating their success on LJspeech and VCTK proved challenging. Voicebox lacks an available implementation, and unofficial samples suggest that VALL-E faces difficulties in generalizing well when trained only on Lj-speech. **We want to emphasize that our primary focus was introducing a novel approach and exploring the benefits of working directly on the waveform.** We acknowledge that with more resources, it could be feasible to model additional phenomena like background noise, potentially leading to improved results.

---

> ### Author Response · Authors · 2023-11-20
> **Response to reviewer a3Eo (part 3/4)**
>
> **Minor**
>
> - We have attached **RTF values** compared to the other models. As mentioned, employing DDPM+AR and working directly on the waveform comes with a cost in terms of synthesis time. Recognizing this tradeoff, we acknowledge the potential for follow-up work to explore methods for reducing processing time. One approach could involve modifying the denoiser architecture to expedite the process. Alternatively, considering more conventional methods, such as DDIM, presents its own set of advantages and disadvantages.
>
>
> | Syntax      | RTF |
> | ----------- | ----------- |
> | Fastspeech2     | 2.79     |
> | ProDiff  | 0.25     |
> | DiffGAN  | 0.06      |
> | GradTTS  | 3.24      |
> | VITS  | 0.09     |
> | Wavegrad2   | 5.14       |
> | DiffAR  | 31.06      |
>
> - As previously highlighted, our article primarily addresses single-speaker synthesis. Nevertheless, we recognize the potential for extending the model to accommodate multiple speakers and varied prosodies. One approach involves leveraging the model's versatility by integrating the speaker's embedding into the synthesis process. **We have explored this option using Titanet embedding and datasets like VCTK/LibriTTS** with detailed insights provided on GIT regarding results and architecture structure. Additionally, a combination of classifier guidance or free classifier guidance presents an alternative method for introducing features such as emotion.

---

> ### Author Response · Authors · 2023-11-20
> **Response to reviewer a3Eo (part 4/4)**
>
> **Questions**
>
> **Q1:**
>
> As mentioned earlier, we acknowledge the possibility of having conducted a more comprehensive comparison with existing TTS models and regret not doing so sooner. We hope that the attached explanation provides a broader perspective on how our work is positioned within the current TTS landscape. It's important to emphasize that the choice of wavegrad2 was not arbitrary; it is an end-to-end model utilizing diffusion, which was published around the same time as VITS. However, due to its parallel signal generation, it has inherent limitations, resulting in significantly lower audio quality compared to our model. Recognizing the importance of an extensive comparison with more advanced models than fastspeech2 and wavegrad2, we have included additional results.
>
> **Q2:**
>
> The training of the model, following the standard procedure for diffusion models, is stable (A diagram of DiffAR training is provided on git). Currently, we have opted not to backpropagate the predictors at this stage, but this remains a possibility.
>
> **Q3:**
>
> Unfortunately, pitch contour cannot be predicted only from the text as it is related to the way the speaker produces the speech, and changes each and every production.  While our intention was to showcase the model's capacity to be conditioned with additional information and control pitch, we were careful not to rely too much on the pitch predictor's performance if it didn't meet our standards. As a result, we introduced DiffAR+P as an option without directly comparing it to other models.
>
> **Q4:**
>
> In the unconditional task meaningful words are rarely produced. Sometimes they are formed by chance or sound like made-up words, (a phenomenon which is also discussed in WaveNet and WaveTacotron). Even with a high-dimensional manifold (1000 ms), a diffusion model trained on a diverse set of words struggles to reproduce them without guidance. The likelihood of generating meaningful words increases when the dataset has a limited vocabulary, with short words that remain untruncated during training and are repeated sufficiently. An example of this can be seen in DiffWave with the SC09 dataset.

---

### Author Response · Authors · 2023-11-22
**General response to all reviewers**

We would like to thank all reviewers for the constructive feedback aimed to improve the quality of the paper. Following the request of the reviewers to conduct a more comprehensive comparison against existing leading alternative methods, **we have updated the paper with additional comparisons to Grad-TTS, VITS, ProDiff, and DiffGAN-TTS.** We hope that these additional results sharpen the article's contribution and uniqueness compared to existing TTS models.

In light of the existing and added comparisons, we'd like to highlight the main features and novelty of DiffAR:

- DiffAR works **directly on the waveform** and hence it is not a vocoder and does not have decoder-vocoder structure. It does not rely on any lossy representation like the mel-spectrogram. This is part of the novelty of the paper.
- As suggested by the reviewers we added comparisons to **Grad-TTS, VITS, ProDiff, and DIffGAN-TTS.** It can be observed that under the MUSHRA evaluation, DiffAR outperforms these methods. Moreover, DiffAR can model phenomena like vocal-fry, which do not appear in other models.
- By leveraging the stochastic nature of the diffusion process and working directly on the waveform, DiffAR achieves **greater diversity** in its results. Compared to the alternative methods, DiffAR excels in capturing the one-to-many relationship, wherein a given text input can be spoken with various intonations and prosodies.
- In terms of RTF values, DiffAR performs lower than the other methods. A more detailed discussion is provided in the paper.

We have updated the introduction, the experiments and the appendices accordingly.

Please refer to the below tables for details:

**Expanded MUSHRA results:**

(Each table is based on a different set of listeners, hence GT is not the same)

| Model      | MUSHRA |
| ----------- | ----------- |
| GT     | 73.7 &plusmn; 2.4       |
| DiffAR 200   | **69.4 &plusmn; 2.5**        |
| DiffAR 1000   | 67.7 &plusmn; 2.6      |
| GRAD-TTS   | 68.5 &plusmn; 2.5      |


| Model      | MUSHRA |
| ----------- | ----------- |
| GT     | 71.2 &plusmn; 2      |
| DiffAR 200   | **69.5 &plusmn; 2.1**       |
| DiffAR 1000   | 68.4 &plusmn; 2.2      |
| DiffGAN-TTS   | 68.0 &plusmn; 2.2      |

| Model      | MUSHRA |
| ----------- | ----------- |
| GT     | 70.0 &plusmn; 2.1     |
| DiffAR 200   | 66.6 &plusmn; 2.4      |
| DiffAR 1000   | **67.5 &plusmn; 2.3**     |
| ProDiff  | 64.4 &plusmn; 2.4      |

| Model      | MUSHRA |
| ----------- | ----------- |
| GT     | 74.9 &plusmn; 2.2    |
| DiffAR 200   | 69.1 &plusmn; 2.2      |
| DiffAR 1000   | **71.5 &plusmn; 2.2**   |
| VITS  | 69.0 &plusmn; 2.3    |

**RTF results:**


| Model      | RTF |
| ----------- | ----------- |
| Fastspeech2     | 2.79     |
| ProDiff  | 0.25     |
| DiffGAN  | 0.06      |
| GradTTS  | 3.24      |
| VITS  | 0.09     |
| Wavegrad2   | 5.14       |
| DiffAR  | 30.06      |

---

### Meta-Review · Area_Chair_6kug · 2023-12-03

**Metareview:**

The authors present an audio waveform synthesizer based on a diffusion process that can be conditioned on previous overlapping output frames so can be used in an autoregressive manner. The main idea in DiffAR is to sequentially condition each frame on a portion of the previously generated one for diffusion noise prediction. An advantage of the proposed solution is  that it can generate long sentences with almost constant GPU memory cost. A sound experimental validation has also been carried out, and the listening tests are convining. Optimization of the  synthesis process has been left to possible future research.

**Justification For Why Not Higher Score:**

Optimization of the  synthesis process is an important aspect that is missing in the current submission.

**Justification For Why Not Lower Score:**

DiffAR is technically sound solution, and continuously it can generate long sentences with almost constant GPU memory cost.

---

### Decision · Program_Chairs · 2024-01-16

Accept (poster)